# Iron-mediated organic matter decomposition in humid soils can counteract protection

Chunmei Chen[1], Steven J. Hall [2], Elizabeth Coward [3] & Aaron Thompson [4✉]

Soil organic matter (SOM) is correlated with reactive iron (Fe) in humid soils, but Fe also promotes SOM decomposition when oxygen ($O_2$) becomes limited. Here we quantify Fe-mediated OM protection vs. decomposition by adding $^{13}C$ dissolved organic matter (DOM) and $^{57}Fe^{II}$ to soil slurries incubated under static or fluctuating $O_2$. We find Fe uniformly protects OM only under static oxic conditions, and only when Fe and DOM are added together: de novo reactive $Fe^{III}$ phases suppress DOM and SOM mineralization by 35 and 47%, respectively. Conversely, adding $^{57}Fe^{II}$ alone increases SOM mineralization by 8% following oxidation to $^{57}Fe^{III}$. Under $O_2$ limitation, de novo reactive $^{57}Fe^{III}$ phases are preferentially reduced, increasing anaerobic mineralization of DOM and SOM by 74% and 32–41%, respectively. Periodic $O_2$ limitation is common in humid soils, so Fe does not intrinsically protect OM; rather reactive Fe phases require their own physiochemical protection to contribute to OM persistence.

[1] Institute of Surface-Earth System Science, Tianjin University, Tianjin 300072, China. [2] Department of Ecology, Evolution, and Organismal Biology, Iowa State University, Ames, IA 50011, USA. [3] Delaware Environmental Institute, University of Delaware, Newark, DE 19711, USA. [4] Department of Crop and Soil Sciences, University of Georgia, Athens, GA 30602, USA. ✉email: AaronT@uga.edu

The net balance of soil carbon (C) accrual vs. loss is central to future climate predictions. Accumulating research has demonstrated that geochemical factors, such as secondary clay minerals and short-range-ordered (SRO) iron (Fe), and aluminum (Al) phases, in particular, are vital determinants of C accrual[1–3]. Mineral-associated organic matter (MAOM) is thought to persist because organic matter (OM) can form strong chemical bonds to minerals and can be physically protected in microaggregates or co-precipitates[4,5]. Once the initial association of OM with minerals has occurred, soil structural conditions (aggregate formation, macro-scale shifts in fluid flowpaths, etc.) can further isolate and compartmentalize OM from decomposer organisms and restrict the diffusion of oxygen ($O_2$), thus further protecting soil organic matter (SOM) against decomposition[6,7]. These features can lead to longer turnover times for MAOM than for particulate organic matter[8,9], and may explain MAOM residence times of centuries–millennia[4,5,10]. A large portion of MAOM in soils and sediments is adsorbed or co-precipitated with Fe minerals[11–13]. However, soil Fe plays multiple roles in ecosystem biogeochemistry aside from C protection, some of which also drive C loss.

Soil Fe serves three categorical roles in ecosystem function (Fig. 1 and Supplementary Fig. 1): the first is a structural role, where Fe (as $Fe^{III}$) forms connective cements that bind minerals and SOM together in nano-, micro-, and macro-aggregates[7,14]; the second is a sorbent role, whereby nutrients and OM adsorb or co-precipitate with $Fe^{III}$ minerals or $Fe^{III}$ surface coatings[5]; and the third is an electron-transfer role, whereby $Fe^{III}$ accepts electrons from microbes or electron shuttles, or $Fe^{II}$ donates electrons to various oxidants, such as $O_2$, $NO_3^-$, or $H_2O_2$[15]. The relative impacts of these Fe functional roles on soil C cycling remain unclear.

The sorbent and structural roles of Fe may increase soil C stocks by decreasing the availability of OM to extracellular enzymes and heterotrophic microbes[5,7]. A commonly accepted mechanism for MAOM formation is for dissolved organic matter (DOM) of plant or microbial origin[16] to sorb or co-precipitate with existing and de novo minerals[5,17–19]. One particularly important route of MAOM formation involves the oxidation of $Fe^{II}$ to $Fe^{III}$ at redox interfaces and its rapid hydrolysis to SRO $Fe^{III}$ (oxyhydr)oxides, which co-precipitate with DOM[20]. This can occur wherever $Fe^{II}$-bearing anoxic solutions come in contact with $O_2$, such as in periodically flooded soil horizons or across redox gradients within aggregates in upland soils[20–22]. High rates of Fe reduction have been observed in surface soils during periods of elevated moisture and high biological activity, leading to a heterogeneous distribution of

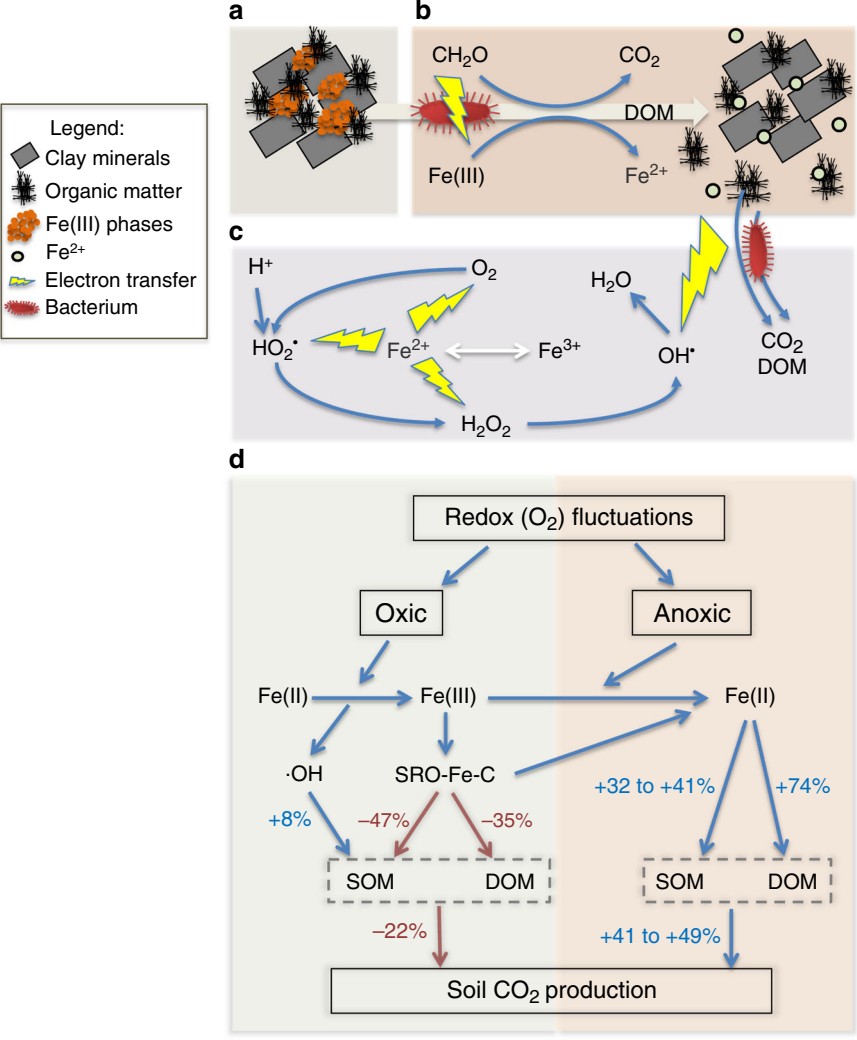

**Fig. 1 Schematic of Fe-mediated C transformation.** Under oxic conditions, $Fe^{III}$ phases sorb C (**a**). Under anoxic conditions, $Fe^{III}$ can be reduced to $Fe^{II}$ coupled with C mineralization, releasing dissolved organic matter (DOM) (**b**). $Fe^{2+}$ oxidation yields reactive $O_2$ species driving $CO_2$ and DOM production (**c**). **d** Conceptual diagram of contrasting positive (blue arrows) and negative (red arrows) changes in $CO_2$ production linked to a given mechanism.

iron within soil profiles[23–26]. Iron reduction appears to be a ubiquitous soil biogeochemical process across a broad range of terrestrial ecosystems[23–30]. Across these ecosystems, C:Fe molar ratios of Fe–C associations point to the dominance of co-precipitation vs. adsorption[11,12]. These lines of evidence place the epicenter of Fe-associated OM formation at these dynamic anoxic-oxic interfaces in surface soils.

However, the biogeochemical factors linked to Fe-associated C formation could also contribute to its decomposition. Fe electron transfer reactions can drive C solubilization, depolymerization, and loss as $CO_2$. During anoxic periods, microbial use of $Fe^{III}$ as an electron acceptor directly produces $CO_2$ from the metabolic coupling of OM oxidation to Fe reduction[27–29], but also releases OM from $Fe^{III}$–OM coprecipitates and OM occluded in $Fe^{III}$-cemented micro-aggregates[30–32]. In soils that experience frequent redox fluctuations, microbial Fe reduction can account for up to 44% of anaerobic OC mineralization[33]. Therefore, significant portions of C protected by complexation under oxic conditions (up to 40% of total soil C[11,13]) can potentially be released and decomposed following Fe reduction. Conversely, the abiotic oxidation of $Fe^{II}$ by $O_2$ can also produce $CO_2$. This is a consequence of reactive oxygen species production (Fenton chemistry), which can directly produce $CO_2$ or cleave organic polymers to increase OM availability[34–36].

Despite evidence for Fe-stimulated decomposition, the common perception of iron's role in SOM has largely focused on Fe-mediated OM protection via adsorption, co-precipitation, or aggregation[5,7,12,19,20,37–40]. While it is also recognized that Fe–OM associations are formed during Fe redox cycling, and that Fe oxidation and reduction can promote C release and mineralization[31–33,36], these processes are rarely explored concurrently. In fact, few studies have directly measured the microbial availability of Fe-associated OM in soils[40,41], and studies highlighting Fe-associated C in anoxic zones do not examine why these $Fe^{III}$ minerals persist despite being thermodynamically poised for reductive dissolution[12,20]—this is a topic of separate studies explaining $Fe^{III}$ stability based on the thermodynamic constraints that OM composition places on $Fe^{III}$ respiration[22,42,43]. Examining these competing functional roles together remains a critical knowledge gap.

In this study, we quantified the relative contributions of Fe in retarding and accelerating C loss in the initial stages of MAOM formation, where physical constraints (macroaggregation, etc.) on decomposition were minimized using soil slurries (Fig. 1). We hypothesized that the electron transfer roles of Fe, which accelerate C mineralization, counteract C protection by Fe's sorbent roles during and shortly following MAOM formation. To test this, we amended soil slurries with $^{57}Fe^{II}$ and/or $^{13}$C-DOM under anoxic conditions and formed Fe–MAOM by introducing $O_2$, simulating a primary mechanism of Fe–MAOM formation in humid soils. The soil slurries were incubated under either static oxic or alternating oxic/anoxic treatments under pH-buffered conditions, and the added and extant Fe and C were tracked using isotope measurements. Comparing treatments with and without added $^{57}$Fe, we find that adding $^{57}Fe^{II}$ only decreases $CO_2$ production when added together with $^{13}$DOC, and only under static oxic conditions. When $^{57}Fe^{II}$ is added alone, Fenton chemistry promotes more SOM decomposition following oxidation of $Fe^{II}$ to $Fe^{III}$ than the additional de novo $^{57}Fe^{III}$ minerals protect existing SOM. In fluctuating redox treatments, the added $^{57}$Fe provides an additional electron acceptor to fuel $CO_2$ production during anoxic periods and $^{13}$DOC trapped by de novo $^{57}Fe^{III}$ phases is released, increasing anaerobic $^{13}CO_2$ production relative to static oxic treatments. This study highlights that Fe's electron transfer roles can largely counteract the protective effect of SRO–$Fe^{III}$-C associations and sustain C decomposition in redox-dynamic systems.

## Results and discussion

**Synopsis.** Consistent with a protective role, under static oxic conditions we found that $Fe^{II}$ oxidation in the presence of added $^{13}$C-DOM resulted in SRO Fe–C associations that not only inhibited the mineralization of $^{13}$C-DOM by 35% relative to controls, but also suppressed the priming of native SOM mineralization by 47%, which consequently decreased overall $CO_2$ production by 22% (Fig. 1d). However, when $^{13}$C-DOM was not added, $Fe^{II}$ oxidation and the production of reactive oxygen species stimulated mineralization of native SOM by 8% relative to the controls (Fig. 1d). Thus, the formation of additional SRO–Fe phases did not provide net protection to SOM unless there was additional DOM present. As might be expected, the protective role of Fe was reversible under anoxic conditions. Although $CO_2$ production from non-Fe amended treatments during the anoxic period was 68–70% lower than in the static oxic treatment (Fig. 1d), the de novo SRO Fe–MAOM formed via $Fe^{II}$ oxidation was disproportionately vulnerable to subsequent reduction. This consequently stimulated the mineralization of both added $^{13}$C-DOM and the native SOM by 74% and 32–41%, respectively, and thus increased overall $CO_2$ production by 41–49% relative to both non-Fe amended treatments (with or without added DOM, Fig. 1d). As a result of Fe-stimulated C mineralization, the anaerobic $^{13}$C-DOM mineralization was 81% greater than the oxic control. Below we provide details on the production of the Fe–MAOM, discuss the data supporting Fe protection of C along with the data supporting Fe stimulation of C loss, and then provide a synthesis of the work.

**Generation of $Fe^{III}$-(oxyhydr)oxides.** The oxidation of $^{57}Fe^{II}$ after a 1-d equilibration with the soil under anoxic conditions generated SRO $Fe^{III}$ (oxyhydr)oxides that impacted C cycling. Exposure to $O_2$ (day 1–6) led to the oxidation of $Fe^{II}$, with aqueous $Fe^{II}$ completely oxidized within 6 h. The sorbed $Fe^{II}$ substantially decreased by 91% over the first day and slowly declined thereafter (Fig. 2). The treatment with both $^{57}$Fe and $^{13}$C-DOM added had 10% more adsorbed $^{57}Fe^{II}$ than the $^{57}Fe^{II}$-only treatment before oxidation (Fig. 2a and b), likely due to co-sorption of the $Fe^{2+}$–DOM complex, as observed previously[44]. The variable-temperature Mössbauer spectroscopy technique that we use to track the mineral composition of the $^{57}$Fe additions, gives excellent information on the crystallinity of the Fe phases, with high crystallinity phases ordering at higher temperatures. Both $^{57}$Fe addition treatments led to the formation of de novo SRO $^{57}Fe^{III}$ phases of lower crystallinity (lower Mössbauer ordering temperature) than the bulk soil Fe (Table 1; Supplementary Fig. 2 and Fig. 3), resulting in a 26–31 mmol kg$^{-1}$ increase in lepidocrocite and 3–14 mmol kg$^{-1}$ increase in nano-goethite and very-disordered $Fe^{III}$ (oxyhydr)oxides that preclude assignment (Fig. 3; Supplementary Table 1). The addition of $^{57}$Fe and $^{13}$C-DOM together resulted in the formation of even lower crystallinity SRO $Fe^{III}$ (oxyhydr)oxides than the $^{57}$Fe addition-only treatment as illustrated by the lower 35K/5K and 12K/5K crystallinity ratios (Table 1; Supplementary Fig. 3). Suppression of $Fe^{III}$ crystallinity by co-precipitation with dissolved fulvic acids has been shown previously in synthetic pure systems[44], and here we extended this finding to a complex soil system containing a mixture of aluminosilicates, $Fe^{III}$ (oxyhydr)oxides, and a variety of organic compounds. In general, lower crystallinity Fe (oxy-hydr)oxides (often measured by oxalate-extraction) have higher surface area, sorb more OM, and are thought to be associated with persistent OM in soils[45].

**Iron protection of organic matter.** In the static oxic treatment, addition of $^{57}$Fe suppressed the mineralization of $^{13}$C-DOM by

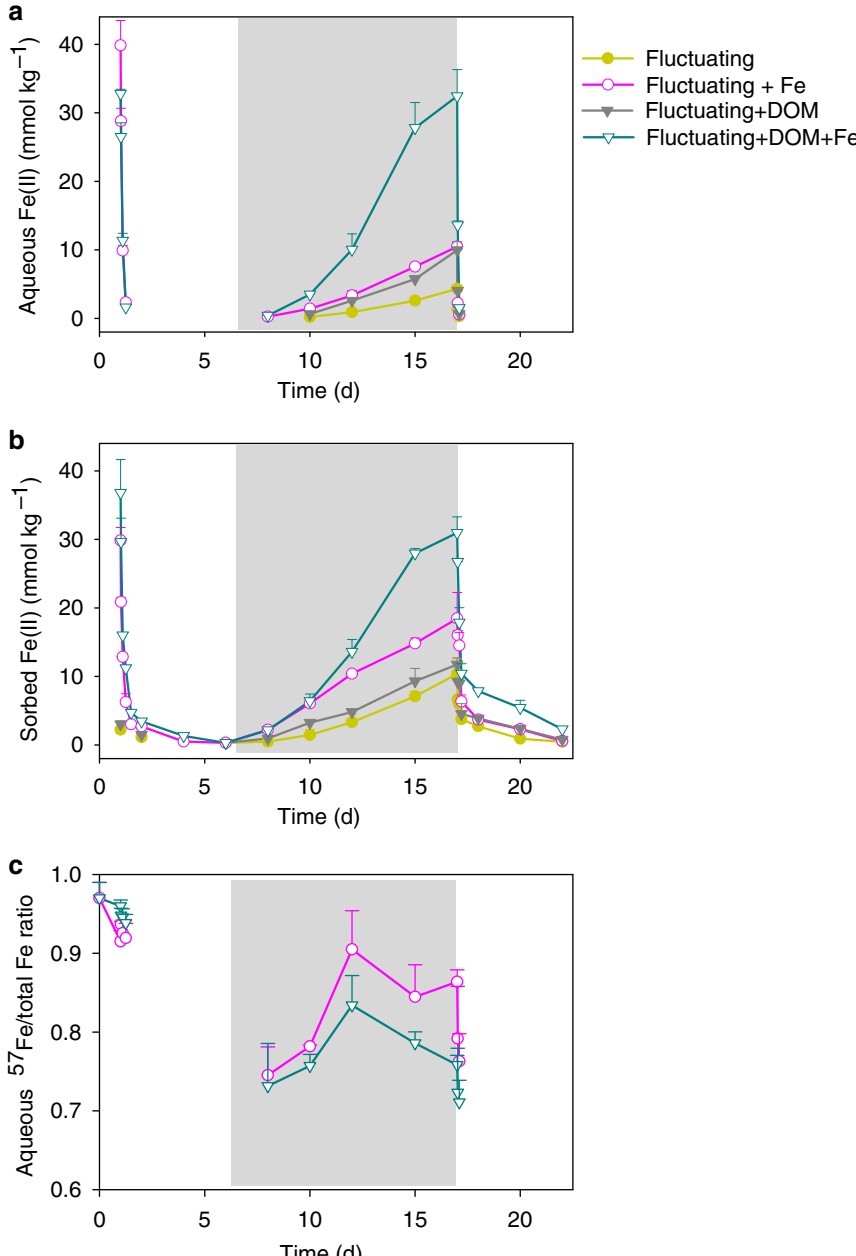

**Fig. 2 Fe^II concentration and ^57Fe to total Fe ratio.** Time-dependent **a** aqueous Fe^II, **b** sorbed (HCl-extractable) Fe^II, and **c** dissolved ^57Fe to total Fe ratio in the aqueous phase in the redox-fluctuating treatment. Fe^II and ^57Fe were undetectable during day 1.5–6 and 17.5–22. The gray shaded region represents the anoxic phase of the redox-fluctuating treatment. Error bars indicate s.e.m. ($n = 3$).

35% ($p < 0.01$, Figs. 1d, 4 and Table 2): cumulative $CO_2$ production was $25.6 \pm 0.8$ and $39.5 \pm 1.1$ mmol C kg$^{-1}$ with and without ^57Fe, respectively, equivalent to 17.1% and 26.3% of the added ^13C-DOM (Fig. 4c and Table 2). Although ^57Fe addition also inhibited net ^13C-DOM mineralization in the fluctuating redox treatments ($p < 0.01$, Fig. 4c and Table 2), this inhibition was confined to the oxic portions (days 1–6 and days 17–22) of the incubation and was partly offset by a 74% enhanced ^13C-DOM mineralization during the anoxic phase (days 6–17) relative to the treatment without added Fe (Figs. 1d, 4b, and Table 2, see below).

The generation of low crystallinity SRO–Fe^III (oxyhydr)oxides from the oxidation of ^57Fe^II in the presence of ^13C-DOM resulted in a lower DOC concentration than in the ^13C-DOM-only treatment and a concurrent increase in solid-phase ^13C content

(Fig. 5). This likely reflects the formation of SRO Fe–C complexes with ^13C-DOM adsorbing or co-precipitating with the newly-formed SRO lepidocrocite and nanogoethite phases (Fig. 3). It is generally assumed that SRO Fe phases contribute to soil C persistence by protecting it against microbial mineralization[5], but few studies have directly measured the bioavailability of Fe-associated OM[39]. Our study provides evidence that de novo formation of SRO Fe–C complexes inhibit the mineralization of fresh DOM inputs to soil. Others have also observed a large decrease in OM decomposition when glucose or fulvic acid sorbed to synthetic Fe minerals (ferrihydrite/goethite) was added to soils, as compared to additions of the free organic compounds[40,41]. The bioavailability of mineral-associated OM is generally thought to be linked to C loadings (e.g., C/Fe ratios), with a maximum adsorption capacity occuring at a C/Fe molar

**Table 1 Relative abundance of magnetically ordered Fe$^{III}$ (oxyhydr)oxides in $^{57}$Fe Mössbauer spectra of the initial unreacted soil and the amended $^{57}$Fe (corrected to exclude the signal from the native soil Fe), as a function of temperature.**

| Treatments | Sample time | Magneticially ordered Fe$^{III}$-(oxyhydr)oxides (%) | | | | Crystallinity index | | |
|---|---|---|---|---|---|---|---|---|
| | | 77 K | 35 K | 12 K | 5 K | 77 K/5 K | 35 K/5 K | 12 K/5 K |
| Initial soil | | 57.7 (2.2) | 66.8 (2.1) | 70.6 (2.4) | 76.4 (3.2) | 0.75 | 0.87 | 0.92 |
| Added $^{57}$Fe | | | | | | | | |
| $^{57}$Fe$^{II}$–only addition | Prior to oxic (1 d) | 8.1 (0.9) | 11.4 (1.0) | 29.2 (1.4) | 41.2 (1.0) | 0.20 | 0.29 | 0.71 |
| | End of oxic (6 d) | | 18.5 (2.4) | 62.4 (4.1) | 84.3 (2.8) | | 0.22 | 0.74 |
| | End of anoxic (17 d) | 3.5 (0.7) | 15.3 (0.9) | 57.9 (3.1) | 79.1 (1.0) | 0.04 | 0.20 | 0.74 |
| | End of 2nd oxic (22 d) | 14.7 (1.7) | 20.8 (1.3) | 65.7 (1.9) | 85.6 (2.0) | 0.17 | 0.24 | 0.77 |
| $^{57}$Fe$^{II}$ and $^{13}$C-DOM addition | Prior to oxic (1 d) | 6.2 (0.8) | 8.7 (0.7) | 16.0 (1.1) | 34.7 (1.9) | 0.18 | 0.25 | 0.46 |
| | End of oxic (6 d) | | 13.1 (0.8) | 36.7 (2.8) | 74.0 (3.4) | | 0.18 | 0.49 |
| | End of anoxic (17 d) | 5.2 (0.7) | 7.9 (0.6) | 21.0 (1.0) | 42.0 (3.1) | 0.12 | 0.19 | 0.50 |
| | End of 2nd oxic (22 d) | 11.7 (0.9) | 15.7 (1.0) | 48.3 (3.4) | 77.8 (2.4) | 0.15 | 0.20 | 0.62 |

Numbers in parenthesis represent standard errors associated with Mössbauer data modeling.

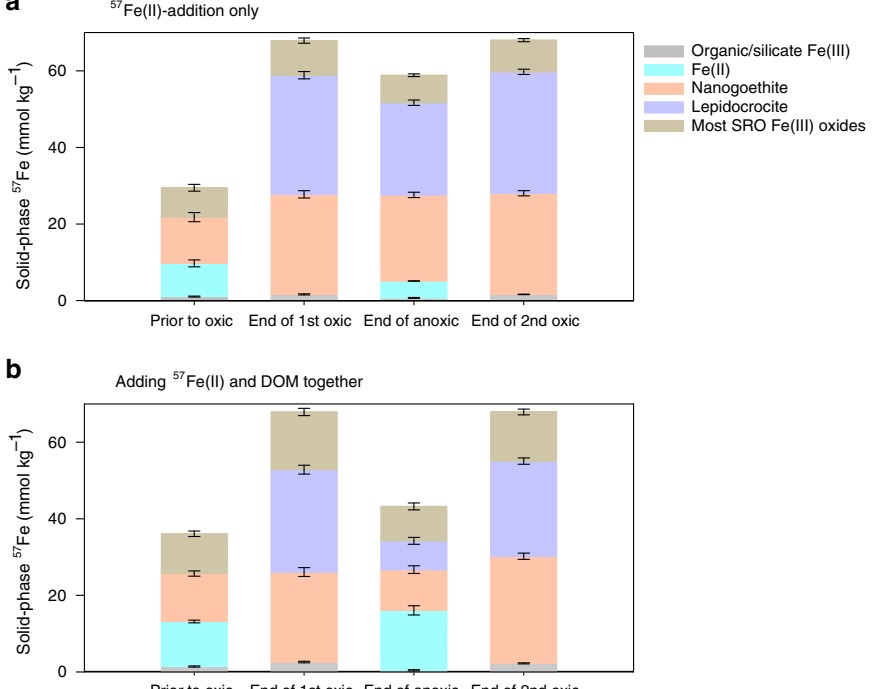

**Fig. 3 Solid-phase speciation of added $^{57}$Fe.** $^{57}$Fe partition was calculated from respective Mössbauer spectra (corrected to exclude the signal from the native soil Fe) for the (**a**) $^{57}$Fe-only and (**b**) $^{13}$C-DOM-$^{57}$Fe addition treatments, prior to the oxic phase (day 1) and at the end of the 1st oxic (day 6), anoxic (day 17) and 2nd oxic (day 22) phases. Error bars represent standard errors associated with Mössbauer data modeling (see Supplementary Information).

ratio of about one[46]. Co-precipitation could result in Fe–OM associations with much higher C/Fe ratios[11,12,19]. In our study, the initial C/Fe molar ratio of the added $^{13}$C-DOM and $^{57}$Fe was 2.1. If we assume that all DOM that was removed from the solution during the Fe$^{II}$ oxidation event sorbed to the newly-formed Fe$^{III}$ (oxyhydr)oxides, the C/Fe ratio of those OM–Fe$^{III}$ (oxyhydr)oxide complexes would be ~1.7. Thus, there was likely $^{13}$C-DOM with a low affinity for Fe$^{III}$ (oxyhydr)oxides that remained as unprotected $^{13}$C-DOM in the aqueous phase and this likely led to our observation of significant $^{13}$C-DOM mineralization even in the presence of de novo Fe$^{III}$ (oxyhydr)oxides (Fig. 4).

Labile C inputs are often observed to alter the decomposition of extant SOM, defined as priming[47,48]. During the oxic periods of the experiment, $^{57}$Fe$^{II}$ oxidation in the presence of added $^{13}$C-DOM

not only suppressed the mineralization of the amended $^{13}$C-DOM, but also partially inhibited the priming of native SOM decomposition compared to the DOM-only treatment (Fig. 6; Table 2; Supplementary Table 2). In the static oxic treatment, addition of $^{13}$C-DOM alone or together with $^{57}$Fe increased native SOM-derived CO$_2$ production compared to the soil-only control (priming effect) ($p < 0.01$, Fig. 6a, c and e; Table 2; Supplementary Table 2). However, adding $^{13}$C-DOM and $^{57}$Fe together resulted in a significantly smaller priming effect on native SOM mineralization than adding $^{13}$C-DOM alone under the static oxic treatment ($p < 0.01$, Fig. 6e, Table 2 and Supplementary Table 2). With the addition of $^{13}$C-DOM, cumulative primed CO$_2$ from native SOM under the static oxic treatment measured $10.2 \pm 1.2$ and $19.1 \pm 0.9$ mmol C kg$^{-1}$ with and without $^{57}$Fe addition, respectively (Fig. 6e and Supplementary Table 2). Cumulatively, adding

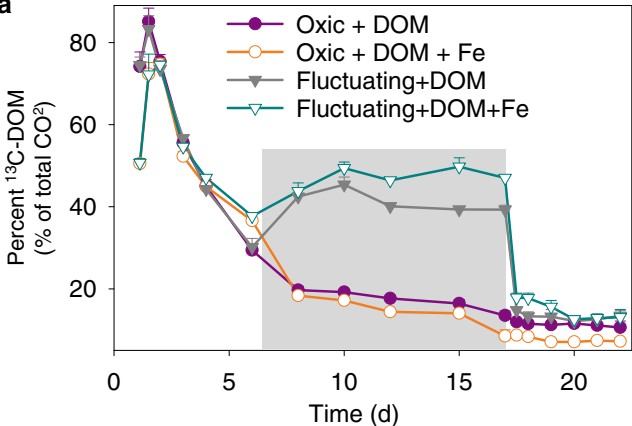

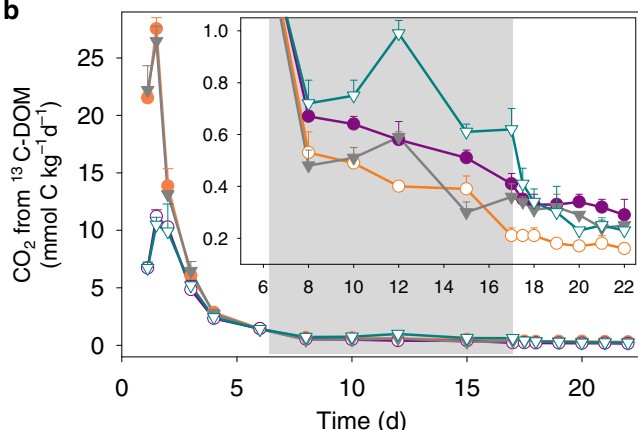

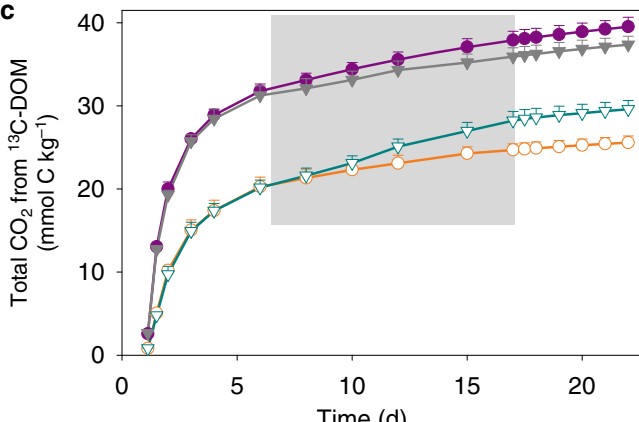

**Fig. 4 Mineralization of the amended $^{13}C$-labeled DOM. a** The percent contribution of $^{13}C$-labeled DOM to total $CO_2$ production, **b** $CO_2$ production rates from $^{13}C$-labeled DOM, and **c** cumulative $CO_2$ production from $^{13}C$-labeled DOM. The gray shaded region represents the anoxic phase of the redox-fluctuating treatment. Error bars indicate s.e.m. ($n = 3$).

$^{13}C$-DOM together with $^{57}Fe$ suppressed aerobic priming of native SOM by 47% relative to adding $^{13}C$-DOM alone (Fig. 1d, Table 2 and Supplementary Table 2). Collectively, $^{57}Fe$ oxidation in the presence of $^{13}C$-DOM resulted in 22% less overall C mineralization, compared to addition of $^{13}C$-DOM alone (Fig. 1d and Table 2).

We propose that de novo SRO Fe$^{III}$ minerals protected the added $^{13}C$-DOM under static oxic conditions, decreasing DOM availability for microbial growth, and suppressing the priming of native SOM. An alternative explanation, that the sorption of

native SOM onto the de novo SRO Fe$^{III}$ (oxyhydr)oxides inhibited priming, is unlikely because when we added $^{57}Fe^{II}$ alone it actually increased the mineralization of native SOM (due to reactive oxygen species production, as discussed below) (Fig. 6a, e and Table 2). Similarly, prior studies have shown that the addition of new SRO Fe$^{III}$ phases to soils has little to no impact on the mineralization of native SOM[41,49]. Rather, it is likely that DOM–Fe$^{III}$ interactions create physico-chemical barriers that limit priming by decreasing microbial access to the new DOM. Thus, when reduced soils receive oxygenated water due to rainfall, snowmelt, or irrigation, the oxidation of Fe$^{II}$ in the presence of DOM and formation of OM–Fe$^{III}$ complexes may contribute to C protection both directly, as previously known, and indirectly, by suppressing priming.

**Iron stimulation of DOM and SOM mineralization.** Only in the treatment where $^{13}C$-DOM and $^{57}Fe$ were added together were we able to confirm that Fe had an overall protective effect on OM, and that protection was limited to the oxic portions of the experiment. Below, we quantified the impact of Fe on OM mineralization via Fe-stimulated Fenton chemistry during the first few days of oxic exposure and via Fe reduction-mediated reactions during the anoxic periods.

Adding $^{57}Fe$ alone strongly stimulated $CO_2$ production from native SOM during the first 3 days of the static oxic treatment (and the oxic portions of the fluctuating redox treatment) relative to the soil-only control ($p < 0.01$), with no stimulatory impact afterwards (Fig. 6a and e). Cumulatively, Fe$^{II}$ oxidation stimulated $CO_2$ production by 8% (Fig. 1d and Table 2). To confirm the role of Fenton chemistry, we performed a parallel experiment with added terephthalate—an effective hydroxyl radical scavenger—and found similar $CO_2$ production between Fe$^{II}$-added treatments and soil-only controls (Supplementary Fig. 4). Recent studies have similarly shown that Fe$^{II}$ oxidation is linked to increases in soil $CO_2$ production via the generation of radical oxygen species[50,51], which facilitate the breakdown of complex biopolymers to produce labile substrates for microbial respiration[34,35]. Others have also attributed increased $CO_2$ production following Fe$^{II}$ oxidation to an increase in acidity that can promote DOC release[36]. Given that we conducted these experiments in a strong buffer at a constant pH, the increased $CO_2$ production following Fe$^{II}$ oxidation was most likely derived from the production of reactive oxygen species such as the hydroxyl radical.

Soil C mineralization rates typically decrease as $O_2$ becomes limiting[22,52,53]. In our soil-only control, $CO_2$ production from native SOM during the anoxic period was 70% lower than in the static oxic treatment (Figs. 1d, 6 and Table 2). However, during the anoxic portions of the experiment, Fe addition stimulated native SOM mineralization relative to the no-Fe treatment (Fig. 6b and f; Table 2). In the $^{57}Fe$-addition treatment, the degree of anoxic suppression of $CO_2$ production decreased from 70 to 58% of that under oxic conditions ($p < 0.05$; Table 2), as a result of a 41% higher anoxic native SOM-derived $CO_2$ production in the Fe addition treatment than in the no-addition control ($p < 0.05$, Table 2; Figs. 1d, 6b, d and f). This likely resulted from enhanced microbial use of Fe$^{III}$ as an electron acceptor in the $^{57}Fe$ addition treatments. Following the transition from oxic to anoxic conditions in the fluctuating redox treatments, substantial Fe$^{III}$ reduction occurred (day 6–17, Fig. 2a and b) and adding $^{57}Fe$ increased the total Fe$^{II}$ production rates (2.9 mmol kg$^{-1}$ d$^{-1}$) compared to the soil-only control (1.6 mmol kg$^{-1}$ d$^{-1}$). This was most likely due to the facile reduction of de novo $^{57}Fe$ SRO lepidocrocite and nanogoethite, which had a much lower crystallinity (and thus a higher reactivity) than native soil Fe$^{III}$ (oxyhydr)oxides (Table 1 and Fig. 3; Supplementary Table 1 and Fig. 3). The

**Table 2 Cumulative CO₂ production in the fluctuating redox and static oxic treatments for all soils.**

| Substrate treatment | Time | Redox treatment | SOM-derived CO₂ (mmol C kg⁻¹) | ¹³C DOM-derived CO₂ (mmol C kg⁻¹) | Total CO₂ (mmol C kg⁻¹) |
|---|---|---|---|---|---|
| Soil-only | 1–6 d | 1st-oxic/fluctuating | 11.4 (0.4) | | 11.4 (0.4) |
| | | Static oxic | 11.3 (0.6) | | 11.3 (0.6) |
| | 6–17 d | Anoxic/fluctuating | 6.6 (0.5) | | 6.6 (0.5) |
| | | Static oxic | 21.8 (1.4) | | 21.8 (1.4) |
| | 17–22 d | 2nd-oxic/fluctuating | 9.8 (0.8) | | 9.8 (0.8) |
| | | Static oxic | 10.1 (0.6) | | 10.1 (0.6) |
| | Sum (1–22 d) | Fluctuating | 27.8 (0.6) | | 27.8 (0.6) |
| | | Static oxic | 43.1 (0.9) | | 43.1 (0.9) |
| Feᴵᴵ-added soils | 1–6 d | 1st-oxic/fluctuating | 14.4 (1.4) | | 14.4 (1.4) |
| | | Static oxic | 14.1 (0.5) | | 14.1 (0.5) |
| | 6–17 d | Anoxic/fluctuating | 9.3 (0.8) | | 9.3 (0.8) |
| | | Static oxic | 22.2 (1.2) | | 22.2 (1.2) |
| | 17–22 d | 2nd-oxic/fluctuating | 9.0 (0.6) | | 9.6 (0.6) |
| | | Static oxic | 10.2 (0.7) | | 10.2 (0.7) |
| | Sum (1–22 d) | Fluctuating | 32.7 (0.9) | | 32.7 (0.9) |
| | | Static oxic | 46.6 (2.1) | | 46.6 (2.1) |
| DOM-added soils | 1–6 d | 1st-oxic/fluctuating | 20.1 (0.9) | 31.2 (1.2) | 51.3 (1.6) |
| | | Static oxic | 20.3 (1.0) | 31.8 (0.9) | 52.1 (2.0) |
| | 6–17 d | Anoxic/fluctuating | 6.7 (0.5) | 4.6 (0.4) | 11.3 (0.9) |
| | | Static oxic | 29.2 (1.3) | 6.1 (0.3) | 35.3 (1.5) |
| | 17–22 d | 2nd-oxic/fluctuating | 9.5 (0.3) | 1.4 (0.1) | 11.1 (0.7) |
| | | Static oxic | 12.8 (0.7) | 1.6 (0.2) | 14.2 (1.0) |
| | Sum (1–22 d) | Fluctuating | 36.3 (0.6) | 37.2 (1.2) | 73.5 (1.7) |
| | | Static oxic | 62.3 (1.1) | 39.5 (1.1) | 101.9 (2.1) |
| DOM- and Feᴵᴵ-added soils | 1–6 d | 1st-oxic/fluctuating | 15.7 (0.7) | 20.2 (0.9) | 35.9 (1.5) |
| | | Static oxic | 16.4 (1.0) | 20.3 (1.1) | 36.7 (2.1) |
| | 6–17 d | Anoxic/fluctuating | 8.9 (0.5) | 8.0 (0.3) | 16.9 (0.9) |
| | | Static oxic | 25.9 (1.8) | 4.4 (0.4) | 30.3 (2.2) |
| | 17–22 d | 2nd-oxic/fluctuating | 8.2 (0.7) | 1.4 (0.1) | 9.6 (0.9) |
| | | Static oxic | 11.1 (0.4) | 0.9 (0.1) | 12.0 (0.8) |
| | Sum (1–22 d) | Fluctuating | 32.8 (0.8) | 29.6 (0.9) | 62.4 (1.7) |
| | | Static oxic | 53.4 (1.1) | 25.6 (0.8) | 79.0 (1.8) |

Numbers in parenthesis represent standard errors ($n = 3$ per treatment).

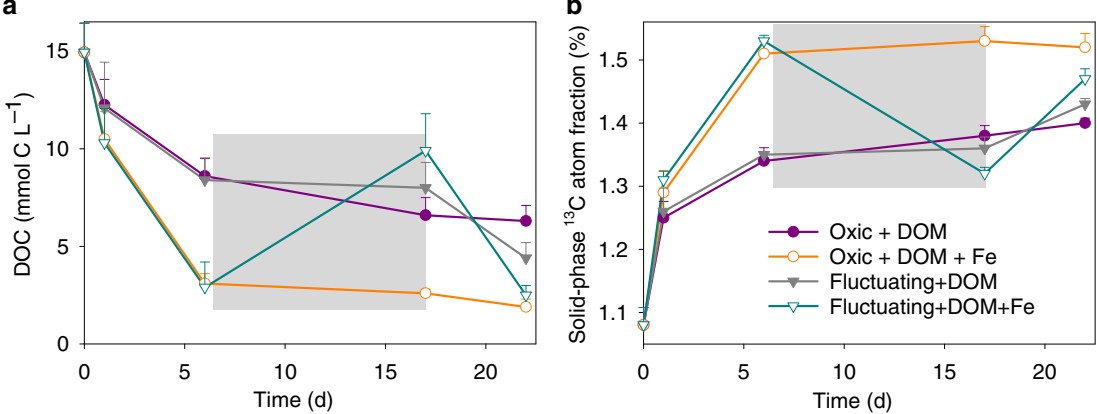

**Fig. 5 Dissolved organic carbon concentration and solid-phase ¹³C enrichment. a** Dissolved organic carbon concentration (MES buffer concentration was subtracted) and **b** solid-phase ¹³C atom fraction from ¹³C-DOM addition treatments. The gray shaded region represents the anoxic phase of the redox-fluctuating treatment. Error bars represent s.e.m. ($n = 3$).

availability of native SRO Feᴵᴵᴵ phases likely limits Fe reduction in this subtropical agricultural soil, and the de novo SRO ⁵⁷Feᴵᴵᴵ phases were preferentially utilized as electron acceptors for microbial respiration as evidenced by the preferential release of ⁵⁷Feᴵᴵ in the aqueous phase (Fig. 2c) and the measured decrease of these ⁵⁷Fe mineral phases following reduction (Fig. 3a and Table 1; Supplementary Table 1).

Iron's stimulation of C mineralization during anoxic periods was greatly enhanced when ⁵⁷Fe and ¹³C-DOM were added together, yielding increases in mineralization of native SOM (anoxic priming) and ¹³C-DOM by $32 \pm 3\%$ and $74 \pm 7\%$, respectively, relative to adding ¹³C-DOM alone ($p < 0.05$; Table 2; Figs. 1d, 4b, 4c, 6b, d and f). In fact, when ¹³C-DOM and ⁵⁷Fe were added together, anaerobic ¹³C-DOM mineralization in the fluctuating redox treatment was

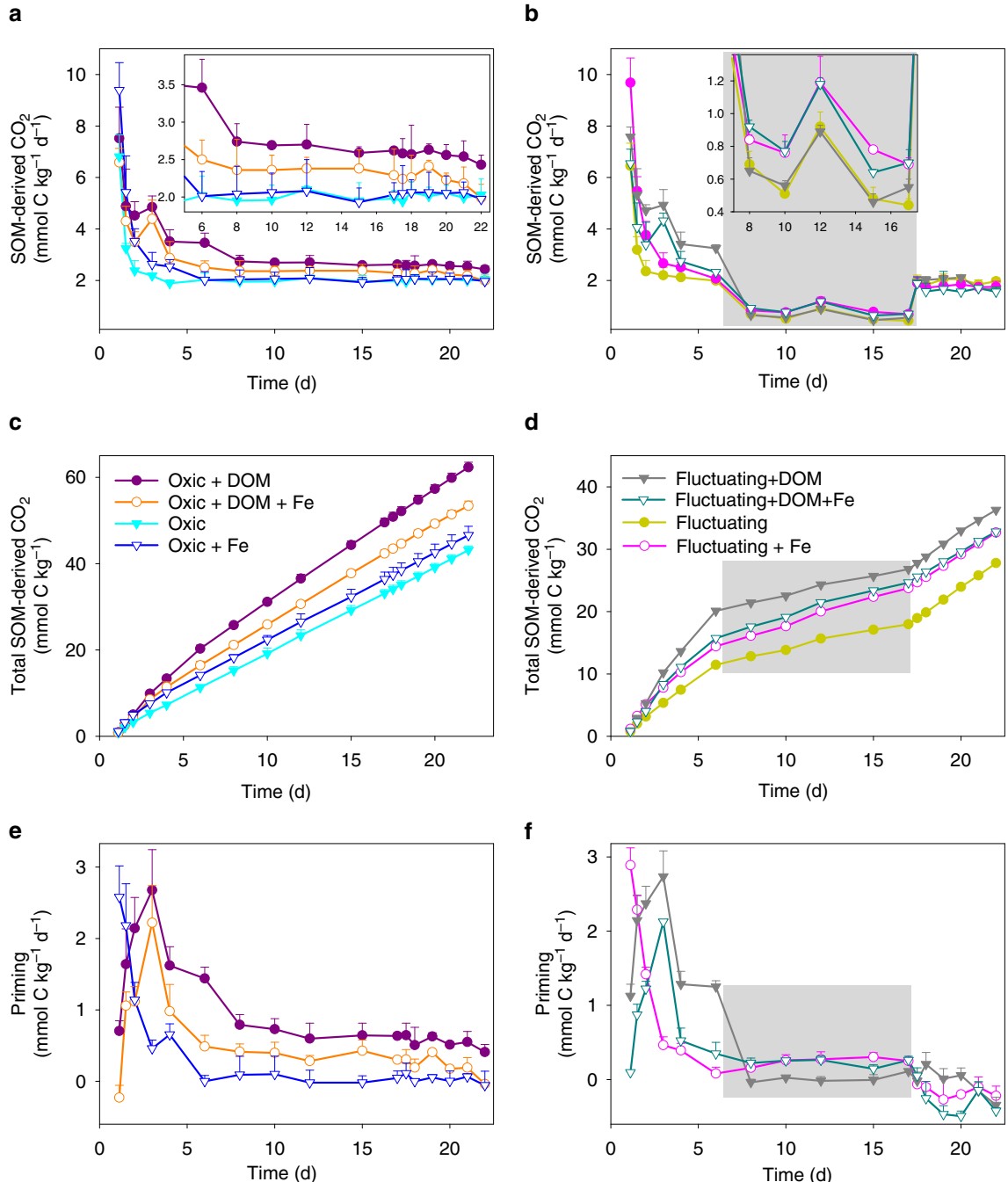

**Fig. 6 Native soil C mineralization.** $CO_2$ production rates from soil C under **a** static-oxic and **b** redox-fluctuating conditions, cumulative $CO_2$ production from soil C under **c** static-oxic and **d** redox-fluctuating conditions, and priming of soil C under **e** static-oxic and **f** redox-fluctuating conditions. The gray shaded region represents the anoxic phase of the redox-fluctuating treatment. Error bars indicate s.e.m. ($n = 3$).

even 81% greater than the aerobic [13]C-DOM mineralization in the static oxic treatment at the same point in time ($p < 0.01$; Fig. 4b; Table 2). The stimulation of [13]C-DOM mineralization under anoxic conditions was linked in part to its molecular composition, given that thermodynamic constraints on Fe reduction limit metabolism to relatively oxidized C substrates[22,42,43]. During the anoxic periods, the mineralization of [13]C-DOM over the native SOM (in both the [13]C-DOM only and DOM–Fe addition treatments) was 2–3 times higher than that in the static oxic treatment at the same point (Fig. 4a). Characterization of the molecular composition of [13]C-DOM and water-extractable native SOM using Fourier transform ion cyclotron resonance mass spectrometry (FTICR-MS) revealed

that the [13]C-DOM had significantly less lignin-derived materials and much more aliphatic formulae than the water-extractable native SOM (Supplementary Table 3, Figs. 5 and 6), which represents the most bioavailable fraction of native SOM[54]. The preferential anaerobic mineralization of [13]C-DOM over SOM may be due to a lower abundance of lignin-derived compounds, which are not readily depolymerized under anoxic conditions[55]. In addition, compared to water-extractable native SOM, [13]C-DOM contains compounds with higher nominal oxidation state of C (NOSC values > 0.5, Supplementary Fig. 6), which are associated with a higher likelihood of thermodynamic favorability ($-\Delta G_r$) when coupled to $Fe^{III}$ reduction than the bioavailable fraction of

native SOM[22,43]. Fe reduction was also stimulated by the addition of [13]C-DOM alone (producing 2.3 mmol kg$^{-1}$ d$^{-1}$ of Fe$^{II}$ compared to the soil-only control rate of 1.6 mmol kg$^{-1}$ d$^{-1}$) (Fig. 2a and b), consistent with prior work[26]. However, when Fe and DOM were added together, Fe reduction was greatly increased to 7.4 mmol kg$^{-1}$ d$^{-1}$, which was even greater than the additive effect of separate [13]C-DOM (2.3 mmol kg$^{-1}$ d$^{-1}$) and [57]Fe additions (2.9 mmol kg$^{-1}$ d$^{-1}$) (Fig. 2a and b). This was because compared to oxidation of [57]Fe$^{II}$ alone, oxidizing [57]Fe$^{II}$ in the presence of [13]C-DOM led to formation of even less-crystalline SRO lepidocrocite and nanogoethite phases (all ordering at <35 K in the Mössbauer spectra, Table 1 and Supplementary Fig. 3). These SRO [57]Fe-[13]C-OM phases exhibited high rates of Fe reduction, releasing significant [57]Fe$^{2+}$(aq) and [13]C-DOM (Figs. 2 and 5) when exposed to anoxic conditions, leaving the solid phase depleted in its lowest crystallinity Fe phases (Fig. 3 and Table 1; Supplementary Table 1 and Fig. 7), and preferentially stimulating anaerobic mineralization of the added [13]C-DOM (Fig. 4a).

Fe reduction can solubilize significant amounts of OM adsorbed or coprecipitated with Fe$^{III}$ (oxyhydr)oxides directly, as shown in our experiment, or indirectly because of an increase in pH[30,32,56]. This re-mobilized [13]C-DOM often includes biochemically labile C[32,57], and may potentially offset the kinetic/thermodynamic constraints often limiting anaerobic decomposition[22,43,58]. We find that collectively, the reduction of SRO Fe$^{III}$ phases offset $O_2$ limitations on C mineralization by 24 ± 3% relative to the non-Fe amended treatment (Table 2).

**Synthesis**. A recent survey of over 5500 soil profiles spanning continental scale environmental gradients found that SRO Fe and Al (oxyhydr)oxide abundance was the best predictor of C content in humid soils, among the geochemical and climate variables that were available[45]. This is consistent with other work showing that SRO Fe$^{III}$ phases are broadly implicated in the persistence of OM in soil[1,3,59]. However, the nature of the relationship between Fe and C in humid soils—and redox dynamic soils in general, which would include floodplain and perennial wetland soils from all climatic regions—is far from straightforward. Humid soils are replete with microsites that undergo dynamic anoxia in response to high labile C loads during periods of high moisture and experience appreciable Fe$^{III}$ reduction rates[23,25,60,61]. Oxidation of the Fe$^{II}$ generated from Fe$^{III}$ reduction is a common mechanism for MAOM formation in humid and redox-dynamic soils, yet Fe is also responsible for OM loss and our work here illustrates two principal refinements in this regard.

First, the production of SRO Fe–MAOM via Fe$^{II}$ oxidation will likely increase $CO_2$ production in the short-term. Only when we formed MAOM in the presence of DOM and maintained strict oxic conditions was there a net decrease in C mineralization (both in the added [13]C-DOM and the native SOM, i.e. via decreased priming). When we simply generated MAOM via Fe$^{II}$ oxidation without added DOM, Fenton chemistry caused an 8% increase in C mineralization (Fig. 1d). Upon the inevitable return to periodic anoxia in humid soils, our work suggests that C mineralization would be accelerated by 41–49% by Fe reduction (Fig. 1d), thus counteracting the stabilization effect on OM of SRO Fe phases. The magnitude of these counteracting mechanisms may also be influenced by soil structure, which we largely eliminated in our study by conducting experiments in soil slurries. Hence, direct application of our results to in situ soil environments is tentative. However, the general principles of our work are also likely to be applicable to structurally complex soil systems. For example, Fe mineral-associated C is often released in natural soils under in-situ flow conditions as a consequence of dissimilatory Fe reduction (e.g.,[62]) and thus becomes more vulnerable to microbial

decomposition. In our study, we even found that the added DOM was preferentially degraded under anoxic conditions relative to the oxic control (Fig. 4), which highlights how the thermodynamic constraints of anaerobic metabolism and the molecular composition of C sources can influence the fate of fresh DOM inputs[22,42,43]. Consequently, the net effect of Fe–C interactions in dynamic redox environments likely hinges in part on the composition of DOM inputs, a worthy topic for further research.

Second, our work here suggests that the initial SRO Fe–C associations are not likely to persist without protection from periodic Fe reduction events. Several researchers have identified or produced SRO–Fe$^{III}$–OM colloids that are resistant to either microbial or chemical reduction[63–67], however, the key components conferring this protection are variable and/or elusive. Some work has identified that SRO–Fe$^{III}$–OM co-precipitates with low C/Fe ratios provide resistance to microbial reduction[63,64], whereas other work has emphasized structural properties (conformation and micro-aggregation) as the mechanism that retards dissolution[65–67]. SRO Fe–OM phases are often co-precipitated with Al and Si ions[68]—which can retard recrystallization[69]—and given the co-association of Al and Fe with OM in humid soils, Al is a strong candidate for protecting Fe against reduction. However, studies that have examined Al and Si co-precipitated Fe-(oxyhydr)oxides found those ions also make the co-precipitates more susceptible to reductive dissolution[70]. Coward et al.[67] recently proposed several mechanisms by which SRO Fe$^{III}$–OM phases could become resistant to reductive dissolution, including acquiring reduction-resistant surface coatings, or becoming embedded in a composite aggregate structure[6]. Such a protective coating could even come from higher crystallinity Fe (oxyhydr)oxides. Hall et al. recently found that [14]C-derived C residence time in humid soils was positively correlated with Fe phase crystallinity[71]. Consistent with that, we find here that in contrast to the initial oxidation event, the 2nd oxidation event generated more crystalline [57]Fe phases (Table 1; Supplementary Fig. 8) and did not stimulate additional C mineralization (Fig. 6 and Table 2). It may be that during repeated redox fluctuations a substantial portion of the co-precipitated OM would be lost, but a core Fe–MAOM structure would remain protected from reductive dissolution.

Perhaps most compelling is the growing evidence that various aggregation, conformation, and structural characteristics of soils confer protection for OM[5–7,10]. Even the protective surface coatings[66,67] or conformational changes in OM at low C/Fe ratios[64] discussed above are examples of micro-aggregate structures not unlike the encasement of SRO Fe–OM phases by aluminosilicate clays or other processes that generate micro-aggregates of minerals and OM during pedogenesis[6,7,10,72]. These aggregation processes can structure microaggregates with core SRO Fe phases and outer aluminosilicate or other phases that are not susceptible to reductive dissolution—as observed in Andisols by dithionite-resistant SRO–Fe phases[66]. Our soil slurry approach was designed to minimize the physical constraints (macro-pore flow, spatial arrangement of microbes, minerals and OM, and the development of aggregates) on C decomposition and thereby isolate the sorbent and electron-transfer roles of Fe in C dynamics (Supplementary Fig. 1). Under these conditions, we find that Fe does not confer intrinsic protection for OM in redox-dynamic soils. In an in situ soil environment— where MAOM emerges in a dynamic three-dimensional space— structural and physical protection of MAOM is thus likely a key protective mechanism for reconciling the comparatively large proportions of SRO-OM associations in soil of very old age based on [14]C-dating[1,4,5,59]. Future studies should thus assess the extent that the formation and destruction of Fe-cemented microaggregates contribute to OM persistence in redox-dynamic soils. Our work demonstrates that the inherent persistence of SRO Fe-associated C cannot be guaranteed. Biological and geochemical context is critical

for understanding the long-term fate of $Fe^{III}$-associated SOM under a changing climate, given the dual roles of $Fe^{III}$ phases in both accelerating and inhibiting OM decomposition.

## Methods

**Approach**. We employed a dual isotope approach in a soil slurry to test our hypothesis that the electron transfer roles of Fe that accelerate C mineralization will counteract C protection by Fe's sorbent roles during, and shortly following, MAOM formation. We used soil slurries (i.e., homogenized mixture of soil and water) to minimize the physical-protection mechanisms of aggregation and the spatial separation of decomposers, substrates, and mineral surfaces, and thus focus on Fe's sorbent and electron-transfer roles. Our dual isotope approach allowed us to distinguish between native SOM and fresh plant-derived DOM via $^{13}C$ labeling, as well as between neo-formed reactive Fe minerals formed in situ and the different forms of Fe minerals in the native soil via $^{57}Fe$ labeling coupled with $^{57}Fe$ Mössbauer spectroscopy.

**Preparation of $^{13}C$-labeled plant-derived DOM**. $^{13}C$-DOM was extracted from $^{13}C$-labeled bermudagrass. DOM is inherently heterogeneous, diverse and dynamic in composition[6], and here we used bermudagrass-extracted DOM to encompass a mixture of organic molecules representative of those that derive from early stage herbaceous litter decomposition. A pulse-labeling method was used to label Tifton-85 bermudagrass (*Cynodon dactylon x Cynodon nlemfuencis*) with $^{13}CO_2$ (99.999 atom%, Cambridge Isotope Laboratories Inc; see Supplementary Methods for additional information). After labeling, aboveground biomass was harvested, immediately frozen, freeze-dried, and then ground using a Wiley mill to <1 mm. DOM extractions were conducted in a shaker at 140 rpm for two days with a solid-to-water ratio of 1:5, followed by centrifugation. The supernatant was filtered through a 0.2 μm membrane filter. The derived DOM solution had 10.3% $^{13}C$. Characterization of the molecular composition using ultrahigh resolution mass spectrometry (FTICR-MS) revealed that this $^{13}C$-enriched DOM was comprised of predominantly aliphatic compounds (76%) and lignin-derived/carboxyl-rich alicyclic molecules (23%), with mean population O/C, H/C and DBE values of 0.44 ± 0.12, 1.60 ± 0.22, and 6.31 ± 3.04, respectively (Supplementary Fig. 6 and Table 3). Compared to water-extractable natural SOM, the bermudagrass-derived DOM had significantly more aliphatic compounds with less lignin-derived materials (Supplementary Notes). In addition, the $^{13}C$-containing population of DOM formulas displayed chemical composition distribution indistinguishable from that of $^{12}C$-only formulae, suggesting no preferential incorporation of $^{13}C$ atoms across molecular compounds (Supplementary Fig. 5 and Table 3).

**Study site and soil sampling**. Our study site is located in the Calhoun Critical Zone Observatory (CZO) in Union County, South Carolina, USA (34.611 N;-81.727, IGSN: IEJCA0013). This site has a humid warm temperate climate, with mean annual precipitation and mean annual temperature of about 1212 mm and 17 °C, respectively (Southeast Regional Climate Center, 2016). The soil used is classified as fine kaolinitic, thermic Typic Kanhapludults of the Appling series, derived from granitic gneiss. We collected soils from cultivated land on an interfluve managed for hay and a few annual crops (e.g., *Zea mais*, *Triticum aestivum*). Current management practice includes annual plowing and disking, the addition of ~4 Mg ha$^{-1}$ of lime in the last eight years and fertilization of NPK at the rate of 160, 40, and 70 kg ha$^{-1}$ yr$^{-1}$, respectively[73]. Interfluves across the Calhoun CZO are characterized by deep soils with pronounced subsurface redoximorphic features[23,74] and seasonal fluctuations in Fe reduction events corresponding with antecedent moisture and labile organic C[75]. During the early spring, surface soils in particular experience a peak in $Fe^{II}$ associated with Fe reduction, which subsequently subsides as the soils become drier later in the spring/summer[75]. Soil pits were dug by backhoe. Surface soils (0–20 cm) were collected and transported overnight to the University of Georgia under ambient conditions. Soils were homogenized and visible plant debris, rocks, and soil macro-fauna were removed manually.

Total OC content and its $\delta^{13}C$ measured via an elemental analyzer-stable isotope ratio mass spectrometer (EA-IRMS) were 2.1% and −22.3‰, respectively. Water-extractable native SOM, extracted by mixing field soils with high purity water (see details in Supplementary Methods), was 33.8 mg C kg$^{-1}$. Water-extractable native SOM was comprised of largely polycyclic aromatic (21.5%), lignin-derived/carboxyl-rich alicyclic molecules (49.1%) and aliphatic compounds (22.5%) with mean population O/C, H/C and DBE values of 0.21 ± 0.09, 1.21 ± 0.35 and 13.73 ± 7.51, respectively (Supplementary Fig. 6 and Table 3). Total Fe content, measured by ICP-MS following Li-metaborate fusion (Acme Labs, Vancouver, BC Canada)[76], was 308 mmol kg$^{-1}$. The concentration of SRO $Fe^{III}$ oxides based on ascorbic acid/citrate extraction[77] was ~25.5 mmol kg$^{-1}$. Soil pH (1:2 ratio of soil: water) was 6.2. XRD analysis revealed that the clay mineralogy of this soil is dominated by kaolinite and illite[78].

**Laboratory incubation**. The experiment had four amendment treatments: soil amended with $^{13}C$-DOM, soil amended with $^{57}Fe^{II}$, soil amended with both $^{13}C$-DOM and $^{57}Fe^{II}$, and control soils with no additions. Each received two redox

treatments: the first $CO_2$-free air (static oxic) treatment; the second treatment with 5 days of $CO_2$-free air, then 11 days of $N_2$ followed by 5 days of $CO_2$-free air again (fluctuating redox treatment). Experiments were performed using soil slurries at a soil:water ratio of 1:10 in triplicate and in the dark under ambient laboratory temperatures of ~24 °C. Field-moist soil (3.5 g, equivalent to 3 g of dry soil) was added to a 125 ml brown amber flask in an anoxic glovebox (Coylabs, Grasslake, MI) with a 95%/5% $N_2/H_2$ atmosphere and stored for 2 h to remove $O_2$, followed by mixing with 30 mL of anoxic MES buffer solution (10 mM, pH = 6). Soil slurries received either 1 ml of anoxic water (controls), or 97 atom% $^{57}Fe$-enriched $Fe^{2+}_{aq}$ added as $FeCl_2$ to reach the initial $Fe^{II}$ concentration of ~70 mmol kg$^{-1}$ soil ($^{57}Fe$-addition treatments). This isotopic enrichment allowed us to monitor the fate of the added $^{57}Fe^{II}$. The added $^{57}Fe$ corresponds to ~91% of total $^{57}Fe$ (added and native) based on native soil $^{57}Fe$ abundance (2.1%), although it is ~18% of the total soil Fe. Anoxic $^{13}C$-DOM solution was added to achieve an initial concentration of 150 mmol C kg$^{-1}$ soil of the added DOM, and thus the added C/Fe ratio is 2.1 in the $^{13}C$-DOM-$^{57}Fe^{II}$ addition treatment. Buffering soil slurries with MES at a constant pH excludes confounding effects of associated pH shifts. The pH of the soil slurries was adjusted to 6 using anoxic HCl or NaOH solutions. The soil slurries were then mixed on a rotary shaker (~250 rpm) in the anoxic glovebox for 1 day to equilibrate the added $^{57}Fe^{II}$ across the aqueous and solid phases under anoxic conditions prior to exposure to $O_2$. Then the reactors were either exposed to static oxic or fluctuating redox treatments by placing the reactors on end-over-end shakers in custom-built, sealed atmospheric chambers (fully contained within the anoxic glovebox) with a continuous flow of either $CO_2$-free air (static oxic treatment) or $CO_2$-free air/$N_2$ alternating treatment. Two sets (3 replicates per set) of parallel samples were prepared: one was used for destructively sampling the soil slurry with the other one reserved for sampling the evolved gas.

To test the effect of hydroxyl radicals on SOM mineralization, 10 mM of terephthalic acid (TPA, an effective hydroxyl radical scavenger) was added to the treatments of soil slurry-only and $Fe^{II}$-amended soil slurry. The reactors were equilibrated under anoxic conditions for 1 d, followed by oxidation with $CO_2$-free air for 5 d. Gas samples were collected for $CO_2$ analysis.

**Soil slurry sampling and analysis**. Sampling of anoxic reactors was performed within the anoxic glovebox. All chemical reagents were prepared in advance with degassed water to preserve Fe oxidation state and for samples collected during anoxic sampling periods. Samples were collected using wide-orifice pipette tips that allowed complete collection of soil particles in the slurry. Aqueous $Fe^{II}$ was extracted from the soil slurries by centrifuging the samples at 14,000 rcf for 10 min, and acid-extractable (sorbed) $Fe^{II}$ was solubilized by suspending the remaining pellet in 0.5 M HCl and shaking it for 2 h on a horizontal shaker at 150 rpm. The extracts were then centrifuged at 14,000 rcf for 10 min and the supernatants analyzed for $Fe^{II}$ using a modified ferrozine protocol[77]. Fe isotope compositions in the aqueous phase and acid-extracts were measured by inductively coupled plasma mass spectrometry (ICP-MS, Perkin Elmer, Elan 9000). Soil slurries were sampled at the end of oxic or anoxic incubations for C and isotope analysis and centrifuged at 14,000 rcf for 10 mins. The supernatant was carefully removed and filtered through a 0.2 μm membrane filter for DOC analysis. DOC was measured with a Shimadzu TOC analyzer. The pellet was washed with anoxic DI water three times, freeze-dried and analyzed for total C and $^{13}C$ analysis using EA-IRMS.

**Gas sampling and measurements**. Each reactor was flushed with $CO_2$-free air or $N_2$ gas for 15 min at 500 mL min$^{-1}$ every 4 h to 1 d during the oxic phases and every 2–3 days during the anoxic period immediately following each headspace gas measurement. We collected gas samples for measurements of $CO_2$ and their $^{13}C$ values immediately prior to flushing, enabling us to quantify cumulative $CO_2$ losses and their $^{13}C$ values over the entire experiment. A 5 ml gas sample was collected with gastight syringes and injected to pre-evacuated 3 mL glass vials (Exetainer, Labco Inc., UK) for $CO_2$ concentration analysis. A 30 ml gas sample was collected from each reactor and stored in helium-purged and evacuated 20-ml glass serum bottles with teflon septa sealed with aluminum crimps for $^{13}C$ measurements. Concentrations of $CO_2$ were measured with a gas chromatograph and thermal conductivity detector (Shimadzu, Kyoto, Japan). Dissolved $CO_2$ in the slurry was calculated based on Henry's law. The $^{13}C/^{12}C$ isotope ratio of $CO_2$ was determined by injecting 20 ml gas using a gas-tight syringe to Piccaro G2201-i via an ultra-zero grade $CO_2$-free air carrier gas. $\delta^{13}C$ values of $CO_2$ from the $^{57}Fe^{II}$-added soils and soil-only controls was corrected using three $CO_2$ tank standards with $\delta^{13}C$ values of −8.0‰, −23.8‰, and −39.7‰ respectively. The $^{13}C$ atom fraction of $CO_2$ from $^{13}C$-DOM-added soils was calibrated using 5 standards varying from 2 to 18% $x$ ($^{13}C$). These standards were created by mixing 99% $x(^{13}C)$ $Na_2CO_3$ with natural abundance $Na_2CO_3$ ($\delta^{13}C = 1.42‰$), digesting with an excess of 12 M HCl and removing aliquots of headspace[79]. Concentrations of $CH_4$ were analyzed by gas chromatography with a flame ionization detector (Shimadzu, Kyoto, Japan). However, $CH_4$ production in this experiment was minimal, accounting for <1% of total C mineralization. Therefore the effect of $CH_4$ production on $^{13}C$ mass balance was negligible[80].

The percent contribution of added $^{13}C$-DOM to $CO_2$ respiration ($P_{DOM}$) was estimated using a two-source mixing model:

$$P_{DOM} = (x^{13}[CO_2]_{D+S} - x^{13}[CO_2]_S)/(x^{13}C_D - x^{13}C_S) * 100$$

where $x^{13}[CO_2]_{D+S}$ and $x^{13}[CO_2]_S$ are atom fraction $^{13}C$ of $CO_2$ respired in the $^{13}C$-DOM amended soils and the treatments with no C addition, respectively; $x^{13}C_D$ is the initial atom fraction $^{13}C$ of $^{13}C$-DOM and $x^{13}C_S$ is the initial soil $^{13}C$. The fraction of $CO_2$ derived from SOM was calculated by difference:

$$P_{soil} = 100 - P_{DOM}$$

Fluxes of $CO_2$ derived from the added DOM and native SOM were calculated by multiplying total $CO_2$ fluxes by their fractional contributions. We calculated priming as the difference in soil-derived $CO_2$ losses between treatments that received $^{13}C$-DOM and/or $^{57}Fe$ additions and soil-only control treatment:

$$C_{primed} = C_{soil\_amended} - C_{soil\_control}$$

**$^{57}Fe$ Mössbauer analysis**. Fe speciation was determined using $^{57}Fe$ Mossbauer analysis. Use of $^{57}Fe$ isotopes allows us to track the amended $^{57}Fe$ using Mössbauer spectroscopy, which detects only $^{57}Fe$ atoms and no other Fe isotopes. The Mössbauer spectra of the amended $^{57}Fe$ was calculated as the difference between the spectra from the $^{57}Fe^{II}$-enriched treatment and the baseline spectra from soils with natural isotopic abundance, after taking into account the different total $^{57}Fe$ concentrations in the $^{57}Fe^{II}$-enriched treatment and the control soils[81]. Therefore, the resulting Mössbauer spectra of the amended $^{57}Fe$ excluded the spectral signal from the native soil Fe atoms (Supplementary Figs. 3, 7–9). To prevent $Fe^{II}$ oxidation, solid samples for $^{57}Fe$ Mössbauer analysis were collected in the anoxic glove box following centrifugation at 14,000 g for 10 min, preserved between layers of $O_2$-impermeable Kapton tape, and immediately frozen in a $-20\,^\circ C$ freezer[81]. The samples were then placed within the spectrometer cryostat (pre-cooled to <140 K), which operated in a He atmosphere to prevent $Fe^{II}$ oxidation by $O_2$. $^{57}Fe$ Mössbauer spectra were recorded in transmission mode with a variable-temperature He-cooled cryostat (Janis Research Co.) and a 1024 channel detector. Detailed information regarding the Mössbauer spectra modeling is provided in the Supplementary Methods. The detailed fitting parameters are presented in Supplementary Tables 4–11.

**Statistical analysis**. A one-way ANOVA (Turkey's HSD) was used to assess the effects of redox treatment on DOM- and SOM-derived $CO_2$ production. A two-way ANOVA was performed to assess effects of DOM and Fe additions on the $CO_2$ production and priming effect. Statistical analysis was performed using SPSS 16.0 for Windows and the differences were considered significant at $p < 0.05$. Iron reduction ($Fe^{II}$ production) rates were calculated from the slope of the linear regression ($R^2 > 0.9$) of $Fe^{II}$ concentration over time during the anoxic period of the fluctuating treatment.

## Data availability
The data that support the findings of this study for all figures are included in a compressed Source Data file accompanying the paper. Other data are included in the Supplementary Materials. Pre-processed data is available upon request.

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

## Acknowledgements

Gratitude is expressed to the National Natural Science Foundation of China (41907013) and the US National Science Foundation (EAR-1331841, EAR-1331846, EAR-1451508, and DEB-1457761) for financial support of the research. We thank Rachel Sleighter for her help with the FTICR-MS analysis.

## Author contributions

C.C. and A.T. conceived of this study. C.C. performed research and analyzed data. E.C carried out FTICR-MS analysis and data interpretation. C.C., A.T., and S.J.H. wrote the paper with the input of E.C.

## Competing interests

The authors declare no competing interests.
