## [Peer Review File · Nature Communications]

Reviewers' comments:

Reviewer #1 (Remarks to the Author):

MANUSCRIPT BRIEF

General Problem

L 24 -25 "Iron (Fe) minerals contribute to the global persistence of soil organic matter (SOM), but they also promote organic matter (OM) decomposition when O₂ becomes limited, and the magnitude of these tradeoffs remains unknown

Specific Research Question

Not explicitly identified

Scientific Unknown

L 54-57 "soil Fe plays multiple roles in ecosystem biogeochemistry aside from C protection, some of which also drive C loss. Quantifying the tradeoffs among these functional roles remains a critical knowledge gap for understanding the persistence or potential losses of Fe-associated C under climate and land-use change."

Objectives

L 112-3 "In this study, we quantify the tradeoffs of C protection and decomposition associated with Fe redox cycling using laboratory incubations that promote the formation of new Fe-C associations as well as their potential decomposition"

Hypotheses

L 114 -117 "we hypothesized that abiotic FeII oxidation would promote OM mineralization, but also form protective SRO FeIII-C complexes that would decrease OM availability under oxic conditions, and that this protection would be lost under anoxic conditions where OM would be released and mineralized"

Conceptual Approach

L 118 -121 "we employed a dual isotope approach (¹³C and ⁵⁷Fe) to simultaneously track the dynamics of C and Fe. Soils were amended with ⁵⁷FeII and/or ¹³C-dissolved organic matter (DOM) to simulate a highly reduced condition, followed by exposure to either static oxic or alternating oxic/anoxic treatments under pH-buffered conditions"

Results

Figure 1 : Cartoon of Fe mediated OM transformations, including panel d) giving "relative changes in CO₂ production linked to a given mechanism"

Figure 2: Aqueous and adsorbed FeII along with ⁵⁷Fe/total Fe ratio as a function of time for 4

"amendment treatments"

Figure 3: Change in the proportions of solid phase Fe compounds over the course of an oxidation/reduction cycle for two out of the four "amendment treatments" (only Fe added, Fe and DOM added together)

Figure 4: Relative contributions of DOM to total CO₂ production, CO₂ production rates, and cumulative CO₂ production for the 4 "amendment treatments"

Figure 5: DOC concentrations and solid phase C enrichment from ¹³C label additions

Figure 6: CO₂ production rates from native SOM in response to various treatments

Table 1: Relative abundance of Fe^{III} oxides from Mossbauer modeling

Table 2: Cumulative CO₂ production for all four "amendment treatments"

ASSESSMENT

This submission is a diligent, elaborate and experimentally very ambitious study of decomposition rates of soil organic matter under fluctuating redox conditions. The focus is on the variable roles that Fe-oxides can play for organic matter decomposition, and an attempt is made at assessing the relative contributions of both retarding and accelerating processes to overall CO₂ production.

The experimental work appears to be solid and the trends observed are plausible, suggesting that the information contained in the manuscript should be of interest to the scientific community.

However, there is a disconnect between the superior quality of the analytical work and the less than satisfying ways the manuscript is organized as a means to inform an audience. For instance, the promise is made to the audience to "quantify the tradeoffs of C protection and decomposition associated with Fe redox cycling".

At this point, the audience needs to be given a clearer understanding what (i.e., which specific experimental observations and outcomes) is meant by the term "tradeoffs". How does one define a "tradeoff" in this specific context and how does one measure a "tradeoff"?

Next we observe a conceptual disconnect between research objective of "quantifying the tradeoffs" and the absence of hypotheses offering estimates of the magnitude of these tradeoffs for testing. The one hypothesis offered "we hypothesized that abiotic Fe^{II} oxidation would promote OM mineralization, but also form protective SRO Fe^{III}-C complexes that would decrease OM availability under oxic conditions, and that this protection would be lost under anoxic conditions where OM would be released and mineralized" is only a very general recapitulation of what most everybody in the community would assume to be true in the first place.

Here, the expectation would be that the authors offer quantitative hypotheses (some a priori assumption of the relative magnitude of the mechanisms considered and tested!) when the declared intent of the authors is to "quantify tradeoffs".

As an audience, we then find ourselves exposed to a quite elaborate experimental design, whose underlying rationale we do not find explained at all. For instance, the use of a label always follows a certain purpose, this purpose should be briefly explained as part of the description of the conceptual approach at the end of the introduction chapter, especially in a situation where such technique is artfully deployed. Capitalize on your strengths!

Once we make our way to the final lines of the manuscript (Synthesis chapter), we find

disappointment: The magnitude of the tradeoffs, the clear objective of this piece, is not synergized into a message statement. Rather, we find ourselves treated to insights such as "if SRO phases are to contribute to OM persistence, they must themselves be protected from reductive dissolution" - but that is a) something we were already aware of and b) it is not the type of message that was promised to us in the introduction.

Curiously, the numbers in Fig 1 panel d) appear to be related to the major objective of this manuscript, yet they are not comprehensively discussed. This reviewer found only two out of a total of 8 of these numbers mentioned in the text (+32 % and + 74 % in line 256), without reference to Figure 1.

RECOMMENDATION

before publication is contemplated, the piece should be revised in two parts:

- a) The Introduction must be harmonized such that a clear logical string leading from question asked over measures taken to answer obtained becomes fully visible.
- b) The Synthesis part must refer back to the question asked and should be dedicated to a clear indication of the type of success achieved. When the promise was to quantify the relative contributions of different mechanisms, then please discuss the implications of the quantitative relationships found and derive an overall insight that may serve as a take-home message for the reader.

Reviewer #2 (Remarks to the Author):

Review of submitted Nature Communications manuscript NCOMMS-19-35427, 27 November 2019

Title: Iron-mediated organic matter decomposition can counteract protection

Authors: Chunmei Chen, Steven J. Hall, Elizabeth Coward, and Aaron Thompson

General Comments

The authors present a study on the interactions between iron oxides and organic matter in a series of soil incubations with labelled dissolved organic matter (^{13}C -DOM) and iron (^{57}Fe), both under oxic conditions and under varying redox conditions (oxic-anoxic-oxic). They find that iron oxides protected native soil and freshly added DOM from remineralization when iron and fresh DOM were added together, as reported in other studies, but also that the addition of highly reactive iron oxides alone (formed from the oxidation of FeII under oxic conditions) lead to enhanced degradation of native soil OM under anoxic conditions owing to their role as electron acceptors. They conclude that iron oxides can protect DOM from degradation only when they are themselves protected from reduction by the co-precipitated DOM. I agree that this conclusion is well supported by the data presented in this work.

This is a very well written, high quality paper with an original experimental approach that allows distinguishing between native soil organic carbon and fresh plant-derived organic carbon through ^{13}C labelling, as well as between neo-formed reactive iron oxides also formed in situ and the different forms of iron minerals in the native soil through ^{57}Fe labelling. The data looks of excellent quality if the standard deviations reported in graphs and tables were obtained from real replicates, as I think they were based on the information provided (but see specific comments below). Mössbauer spectroscopy data brings an interesting layer of complementary information on mineral crystallinity allowing the authors to propose a mechanistic explanation for the bulk and isotopic data derived from the incubation experiment.

My only concerns mostly have to do with additional information that I feel is missing, or explanations

that I believe should be provided. None of these concerns are major however, and I believe that the manuscript should be accepted with minor modifications as it meets the high-quality criteria expected from a publication in Nature Communications. This is an important contribution to the field not so much because of the conclusion on the existence of a mutual protection mechanism between DOM and iron oxides, but because of the elegance of the approach taken to demonstrate the validity of this hypothesis.

Specific comments

1. P. 1, Title: I suggest modifying the title of the manuscript to better reflect the fact that the study focused on soil organic matter: "Iron-mediated organic matter decomposition in soils can counteract protection".
2. P. 6, line 125: Since the Methods section comes later in the paper, I suggest adding a short contextualization sentence here before immediately jumping to the description of the results.
3. P. 8, line 171-172: The sorption capacity of about 1.0 (C/Fe ratio), as presented in your reference #41, corresponds to sorption on the surface of preformed FeIII oxides. Capacities can be higher when FeIII oxides are precipitated from the oxidation of FeII in the presence of DOC, and therefore the ¹³C-DOM remaining in solution might be DOM with a low affinity for the FeIII oxides rather than ¹³C-DOM in excess.
4. P. 11, lines 235-236: The differences in SOM remineralization between the no DOM/no FeII and the no DOM/with FeII treatments as shown on Figure 6b is extremely small and does not appear statistically significant. The claim of a significant difference between the two treatments seem to stem from the numbers in Table 2, where the SOM-derived CO₂ production numbers seem to have been obtained from the sum of the five measurements carried out during the 11-day period under anoxic conditions. If this is correct, then I doubt that the cumulative uncertainty appearing in Table 2 reflects the propagated uncertainty from the 5 individual measurements, or that the replicate measurements are real replicates (one measurement for each one of the 3 replicate (line 401) incubations rather than three measurements on a single incubation experiment). If they are propagated uncertainties on real replicate measurements, I must congratulate you on the quality of the data acquired in this work; if they are not (here and elsewhere), this should be clarified.
5. P. 12, line 256: Please provide the uncertainty on these percentages.
6. P. 12, lines 263-272: Please comment on the possibility that the difference in chemical composition between the water-extractable soil DOM (native DOM) and the fresh, ¹³C-labelled DOM could have influenced the remineralization results during the anoxic portion of the incubation experiment. Ideally, DOM of similar chemical composition but different isotopic composition should have been used, although I realize how difficult to obtain these samples would be – maybe water extractable DOM from agricultural soils where C₄ v. C₃ crops have been grown for a few years?
7. P. 13, line 293: Please provide the uncertainty on this percentage.
8. P. 29, Table 1: Are all these iron oxide minerals really "oxyhydr"oxides or simply hydroxides or oxides?
9. P. S5, 2nd paragraph: The values measured by ICP-MS for the IRMM-014 standard material are very different from the certified ones, particularly for ⁵⁴Fe (4.56±0.11 vs. 5.84±0.02%, respectively) and ⁵⁶Fe (93.12±0.22 vs. 91.75±0.02%, respectively), reflecting the poor accuracy of isotopes ratios determined by ICP-MS. Did you verify that the offset measured for ⁵⁴Fe and ⁵⁶Fe in pure standard solutions remained the same for the sample solutions with a complex matrix? Please comment.

10. P. S5, 2nd paragraph: What were the DOC recoveries in the desalination step? Recoveries can vary a lot from sample to sample when using the PPL cartridges.

11. P. S12, 1st paragraph: What was the range of relative intensities of the subtracted background (^{57}Fe signal in the native soils) relative to the intensities of the spikes? Please provide additional information to allow the reader to appreciate the magnitude of the background correction.

Sincerely,

Reviewer #3 (Remarks to the Author):

The authors present a paper on the role of synthetic iron-oxides in the decomposition of dissolved organic matter (DOM) in slurries. Experiments were run under controlled redox conditions employing completely mixed batch reactor systems with aqueous suspensions (slurries) that were intensively shaken. The suspensions contained labelled synthetic iron oxides and labeled DOM produced within a labeling experiment with Bermudagrass.

General comments: Abstract, Introduction and synthesis

The authors claim to challenge the OM protection “paradigm”, which was developed originally and further elaborated for soils. The experimental results used to justify their proposition are based on an experimental approach, i.e. completely mixed/shaken batch reactor studies, that does not compare with the “normal” situation of organic matter storage and sequestration of organic carbon in terrestrial soils. The processes of OM sequestration are quite diverse and incorporate a vast variety of physical, chemical and biological mechanisms that accompany each other during pedogenesis. Of importance and well established is the role of biota (plant roots, earthworms, earth dwelling insects, microbes, fungi). The interplay of the physical, chemical and biological processes results in an accrual of organic matter over time and thus in a positive OC budget for most of the soil orders. Protection from respiratory or reductive degradation, which factual in soils, is mediated by the processes of structure formation which results in an explicit, soil group specific structural arrangement of the components that build up the soil architecture.

While I strongly respect the research methodology and results obtained, the double labeling approach and the Mössbauer-spectroscopy, I urge the authors to discuss their results in view of pedogenesis by respecting much more the pertinent environmental conditions and the structural arrangement of components in soil (Already tentatively done so in the synthesis section). In addition, the authors should discuss their experimental approach (Adding of DOM and Fe in a slurry) in view of the redox-driven Fe-dynamics in soil: What would be the source of reduced iron required to allow for the formation of Fe-OM in the natural situation where iron is not added but replenished by weathering of Fe-bearing minerals? What about the action of soil biota?

SRO phases of Iron, Aluminum, and Manganese newly formed in the soil environment are protected by being incorporated in the soils aggregate systems, here predominantly as organo-mineral or mineral-organic associations within (micro)aggregates. While the existence of the hierarchy of a system of aggregates is still in debate, aggregation and soils aggregated structure is not disputed. It is also well established that SRO-OM associations are part of the aggregated structure of soils. Aggregation and the emerging architecture of soils is the searched for feature that provides the protection. Or, to state this in alternative phrase – already given by the authors but as a conclusive remark: The processes of soil aggregation provide the protection not only of the SRO Fe phases, but of the associations build of SRO-Fe/Al/Mn and organic matter.

From the results obtained by the authors I would rather conclude that the fact that we find comparatively large proportions of SRO-OM associations in soil of very old age (based on C14-dating

of the OM of the associations) strengthens the concept of protection by aggregation. Given the findings of the authors and others on the sensitiveness of SRO-Fe to a rather rapid decomposition in the absence Oxygen or reductive conditions, the already proposed process of protection in soil aggregates puts the SRO-Fe-OM stabilization paradigm in a greater framework.

Specific comments:

Adding Fe to anoxic conditions may also result, after reduction to ferrous iron, in the formation of Fe-complexes with low-molecular weight DOM. Whether or not this may occur in the experiments and to what extent the results are affected remains unconsidered. Please comment.

L49: There is ample evidence that also secondary clay minerals formed during pedogenesis are important for the accrual of OM in soil. Such secondary clay minerals are not considered in the study of Rasmussen et al. 2018. The base their statement of the "clay content" which is the operational fraction of soil components that are smaller than $<2\mu\text{m}$!

L59-60: The consequence of this "structural role" is the protection of OM against decomposition. It is this structural role that is the basis of the paradigm of OM protection by pedogenic oxides. This should must be emphasized and discussed in the frame of the study.

L61-62: Most of the studies so far concentrated on experimental approaches focusing on sorption/coprecipitation from suspensions/slurries including the work of the authors. A "paradigm" chellenign approach should explore more realistic conditions typical for soils. Soils are no slurries or suspensions.

L75: "ubiquitous" is an overstatement. Based on the two given references, it is not possible to elude on the "ubiquitines" of anoxic microsities in soils. While there is no doubt on the existence of anoxic sites, there is no systematic study on the proportion and persistence of such sites in terrestrial soil orders. In contrast to soils with hydromorphic features (semi-terrestrial soil orders), anoxic conditions are limited to the interior of aggregates that support the respective water retention characteristics/hydraulic properties.

L78-80. There are several studies that specifically explore the reduction/decomposition of Fe-OM and report on the impact on the release of both Fe and DOM as a function.

L85-87: Destruction of the Fe-cemented aggregates by biologic processes and concomitant degradation of OM is one of the processes presumable counteracting OM protection. A better understanding of the mechanisms behind these processes under the conditions met would much better help to test the Fe-OM-stabilization paradigm.

L99-110: This is a truncated and oversimplified restitution of the "paradigm". The full story reads that secondary SRO-Fe/Al-phases show a high affinity for DOM and form associations in soil. However, it is also recognized that they serve as a good substrate and electron donor/acceptor source for microbes and will decompose if not other processes provide protection. Numerous studied showed this under various oxic and anoxic conditions using batch and column techniques. The missing part to understand protection is structure formation and aggregation. Aggregation is the major pathway that results in (physical) protection. The authors should take this part of the "paradigm" into account and revise the MS in accordance.

L117-119: Dual-isotope utilization is a very appropriate approach for such a study. I liked this very much!

L144: The slurry does not resemble the situation of a "complex natural soil". Rephrase.

L180: Why do you classify carbon input by soil biota as "exogenous".

L251, 284: Have you considered the formation of complexes build from ferrous iron and DOM? Such mechanisms have been reported in the literature and may have to be considered in the experimental outcomes. In a natural soil, such complexes would be eventually prone to export with seepage in humid climates. Please comment.

Methods

L332ff: Why did you use Bermudagrass for the labeling experiment? How does Bermudagrass compare to flora of the agricultural sites used for sampling the cultivated soil (L354-355). Is there a bias in OM

quality to be expected?

L339ff: Based on the FTICR-MS: How does the quality of the bermudagrass OM compare to natural DOM <0,2µm? Give details

L359: More details on the soil materials used and the soil used should be given. The Redoximorphic features would much more point to the direction of udults? Ultisols have a broad range of properties, usually strongly acidic and rich in hardly dissolvable Fe-Ox. Cultivation requires melioration of nutrients and H by adding fertilizer and lime. This, of course will affect the type and conditions of the linkage of Fe-OM.

L359ff: Soil material was collected from the first 0-20 cm (topsoil). Are these topsoil horizons known for expressing redoximorphic features?

L394ff: Rotary and end-over-end shaking will further destroy aggregates by mechanical stress. How can you exclude that the results are to a larger extent affected by shaking?

L439ff: Henrys law is not directly applicable to slurries. Did you correct for that?

Reviewers' comments:

Reviewer #1 (Remarks to the Author):

MANUSCRIPT BRIEF

General Problem

L 24 -25 "Iron (Fe) minerals contribute to the global persistence of soil organic matter (SOM), but they also promote organic matter (OM) decomposition when O₂ becomes limited, and the magnitude of these tradeoffs remains unknown

Specific Research Question

Not explicitly identified

Scientific Unknown

L 54-57 "soil Fe plays multiple roles in ecosystem biogeochemistry aside from C protection, some of which also drive C loss. Quantifying the tradeoffs among these functional roles remains a critical knowledge gap for understanding the persistence or potential losses of Fe-associated C under climate and land-use change."

Objectives

L 112-3 "In this study, we quantify the tradeoffs of C protection and decomposition associated with Fe redox cycling using laboratory incubations that promote the formation

of new Fe-C associations as well as their potential decomposition"

Hypotheses

L 114 -117 "we hypothesized that abiotic FeII oxidation would promote OM mineralization, but also form protective SRO FeIII-C complexes that would decrease OM availability under oxic conditions, and that this protection would be lost under anoxic conditions where OM would be released and mineralized"

Conceptual Approach

L 118 -121 "we employed a dual isotope approach (^{13}C and ^{57}Fe) to simultaneously track the dynamics of C and Fe. Soils were amended with $^{57}\text{FeII}$ and/or ^{13}C -dissolved organic matter (DOM) to simulate a highly reduced condition, followed by exposure to either static oxic or alternating oxic/anoxic treatments under pH-buffered conditions"

Results

Figure 1 : Cartoon of Fe mediated OM transformations, including panel d) giving "relative changes in CO_2 production linked to a given mechanism"

Figure 2: Aqueous and adsorbed FeII along with $^{57}\text{Fe}/\text{total Fe}$ ratio as a function of time for 4 "amendment treatments"

Figure 3: Change in the proportions of solid phase Fe compounds over the course of an oxidation/reduction cycle for two out of the four "amendment treatments" (only Fe added, Fe and DOM added together)

Figure 4: Relative contributions of DOM to total CO_2 production, CO_2 production rates , and cumulative CO_2 production for the 4 "amendment treatments"

Figure 5: DOC concentrations and solid phase C enrichment from ^{13}C label additions

Figure 6: CO_2 production rates from native SOM in response to various treatments

Table 1: Relative abundance of FeIII oxides from Mossbauer modeling

Table 2: Cumulative CO_2 production for all four "amendment treatments"

ASSESSMENT

This submission is a diligent, elaborate and experimentally very ambitious study of decomposition rates of soil organic matter under fluctuating redox conditions. The focus is on the variable roles that Fe-oxides can play for organic matter decomposition, and an attempt is made at assessing the relative contributions of both retarding and accelerating processes to overall CO_2 production.

The experimental work appears to be solid and the trends observed are plausible, suggesting that the information contained in the manuscript should be of interest to the scientific community.

Response: Many thanks for your positive comments

However, there is a disconnect between the superior quality of the analytical work and the less than satisfying ways the manuscript is organized as a means to inform an audience. For instance, the promise is made to the audience to "quantify the tradeoffs of C protection and decomposition associated with Fe redox cycling".

At this point, the audience needs to be given a clearer understanding what (i.e., which specific experimental observations and outcomes) is meant by the term "tradeoffs". How does one define a "tradeoff" in this specific context and how does one measure a "tradeoff"?

Next we observe a conceptual disconnect between research objective of "quantifying the tradeoffs" and the absence of hypotheses offering estimates of the magnitude of these tradeoffs for testing. The one hypothesis offered "we hypothesized that abiotic FeII oxidation would promote OM mineralization, but also form protective SRO FeIII-C complexes that would decrease OM availability under oxic conditions, and that this protection would be lost under anoxic conditions where OM would be released and mineralized" is only a very general recapitulation of what most everybody in the community would assume to be true in the first place.

Here, the expectation would be that the authors offer quantitative hypotheses (some a priori assumption of the relative magnitude of the mechanisms considered and tested!) when the declared intent of the authors is to "quantify tradeoffs".

Response: we are very grateful to the reviewer for the insightful comments on our manuscript. These comments have guided us to have a thorough revision on our manuscript, particularly on the section of *Introduction* and *Synthesis*. Following the reviewer's comments, we have re-constructed the section of *Introduction* and setup a framework in a more deliberate manner in the revised manuscript:

First, we clearly clarified the knowledge gap that needs to be addressed, please see P7L140-154: "Despite evidence for the importance of Fe-stimulated decomposition, the dominant paradigm in Earth System science has focused on Fe-mediated OM protection via adsorption, co-precipitation or aggregation^{5, 7, 12, 19, 20, 37-40}. While it is also recognized that Fe-OM associations are formed during redox-cycling of Fe, and that both oxidation and reduction of Fe can lead to C loss and mineralization^{31-33, 36}, these processes are rarely explored concurrently. In fact, only a few studies have directly measured the microbial availability of Fe-associated OM in soils^{40, 41}, and studies that highlight Fe-associated C in anoxic or redox-interface sediments do not explore why these Fe^{III} minerals persist despite being thermodynamically poised for reductive dissolution^{12, 20} — this topic is instead discussed in separate studies highlighting that Fe mineral stability in anoxic zones derives from thermodynamic constraints on decomposition of OM substrates^{22, 42, 43}. Examining these competing functional roles together remains a critical knowledge gap for understanding the persistence or potential losses of Fe-associated C under climate and land-use change."

Second, to avoid the confusion by “quantify the tradeoffs of C protection and decomposition associated with Fe redox cycling”, in the revised *Introduction*, we deleted this statement and focus on a clearer objective (see P8L164-166): “In this study, we sought to quantify the relative contributions of Fe functional roles in retarding and accelerating C loss in the initial stages of MAOM formation, where physical constraints (macroaggregation, etc.) on decomposition were minimized (Fig.1).” In addition, we framed a quantitative hypothesis (see P8L167-169) as the reviewer suggested: “We hypothesized that the electron transfer roles of Fe that accelerate C mineralization will counteract C protection by Fe’s sorbent roles during and shortly following MAOM formation.”

As an audience, we then find ourselves exposed to a quite elaborate experimental design, whose underlying rationale we do not find explained at all. For instance, the use of a label always follows a certain purpose, this purpose should be briefly explained as part of the description of the conceptual approach at the end of the introduction chapter, especially in a situation where such technique is artfully deployed. Capitalize on your strengths!

Response: Thanks for your valuable comments. We have added the rationale of our novel experimental design to test our hypothesis in the revised *Introduction*. Please see P8L169-180:

“To test this, we employed a novel dual isotope approach (^{13}C and ^{57}Fe) to simultaneously track the dynamics of recently formed Fe-C associations. Soils were amended with $^{57}\text{Fe}^{\text{II}}$ and/or ^{13}C - DOM under anoxic conditions and then Fe- MAOM was formed by introducing O_2 , simulating a primary mechanism of Fe-MAOM formation in humid soils. Then the soils were either incubated under static oxic or alternating oxic/anoxic treatments under pH-buffered conditions. We performed the experiments in a soil slurry to minimize the physical-protection mechanisms of aggregation and the spatial separation of decomposers, substrates, and mineral surfaces, and thus focus on Fe’s sorbent and electron-transfer roles. The dual isotope approach allowed us to distinguish between native SOM and fresh plant-derived DOM via ^{13}C labeling, as well as between

neo-formed reactive Fe minerals formed *in situ* and the different forms of Fe minerals in the native soil via ^{57}Fe labeling coupled with ^{57}Fe Mössbauer spectroscopy.”

Once we make our way to the final lines of the manuscript (Synthesis chapter), we find disappointment: The magnitude of the tradeoffs, the clear objective of this piece, is not synergized into a message statement. Rather, we find ourselves treated to insights such as "if SRO phases are to contribute to OM persistence, they must themselves be protected from reductive dissolution" - but that is a) something we were already aware of and b) it is not the type of message that was promised to us in the introduction.

Response: Thank you for insightful comments. In combination with the comments from Reviewer#3, we re-wrote the *Synthesis* section by referring back to our objective that was to quantify the relative contributions of Fe functional roles in retarding and accelerating C loss. In the revised *Synthesis*, we extensively discussed the implications of the quantitative relationships found in our study and derive an overall insight that could serve as a take-home message for the reader, please see P20-23:

“A recent survey of over 5,500 soil profiles spanning continental scale environmental gradients found that SRO Fe and Al (oxyhydr)oxide abundance was the best predictor of C content in humid soils⁴⁵. This dovetails with other work showing SRO Fe^{III} phases are broadly implicated in the persistence of OM in soil^{1, 3, 45, 59}. However, the nature of the relationship between Fe and C in humid soils is far from straightforward. Humid soils are replete with microsites that undergo dynamic anoxia in response to high labile C loads during periods of high moisture and experience appreciable Fe^{III} reduction rates^{23, 25, 60, 61}. Oxidation of the Fe^{II} generated from Fe^{III} reduction is a common mechanism for MAOM formation in humid soils, yet Fe is also responsible for OM loss and our work here illustrates two principal refinements in this regard.

(1) The production of SRO Fe-MAOM via Fe^{II} oxidation will likely increase CO₂ production in the short-term. Only when we formed MAOM in the presence of DOM and maintained strict oxic conditions was there a net decrease in C mineralization (both in the added ¹³C-DOM and the native SOM, i.e. via decreased priming). When we simply generated MAOM via Fe^{II} oxidation without added DOM, Fenton chemistry caused an 8%

increase in C mineralization (Fig. 1d). Upon the inevitable return to periodic anoxia in humid soils, our work shows C mineralization would be accelerated by 41 – 49% by Fe reduction (Fig.1d), thus counteracting the stabilization effect on OM of SRO Fe phases. In our study, we even found that the added DOM was preferentially degraded under anoxic conditions relative to the oxic control (Fig.4), which highlights how the thermodynamic constraints of anaerobic metabolism and the molecular composition of C sources can influence the fate of fresh DOM inputs^{22, 42, 43}. Consequently, the net effect of Fe-C interactions in dynamic redox environments likely hinges in part on the composition of DOM inputs, a worthy topic for further research.

(2) Our work here suggests that, in much the same way wood or steel in a building needs to be painted or encased to protect them from decay and corrosion, the initial SRO Fe-C associations are not likely to persist without protection from periodic Fe reduction events. Several researchers have identified or produced SRO Fe^{III}-OM colloids that are resistant to either microbial or chemical reduction⁶²⁻⁶⁶, however the key components conferring this protection are variable and/or elusive. Some work has identified that SRO Fe^{III}-OM co-precipitates with low C/Fe ratios provide resistance to microbial reduction^{62, 63}, whereas other work has emphasized structural properties (conformation and micro-aggregation) as the mechanism that retards dissolution⁶⁴⁻⁶⁶. SRO Fe-OM phases are often co-precipitated with Al and Si ions⁶⁷—which can retard recrystallization⁶⁸—and given the co-association of Al and Fe with OM in humid soils, Al is a strong candidate for protecting Fe against reduction. However, studies that have examined Al and Si co-precipitated Fe-(oxyhydr)oxides find those ions also make the co-precipitates more susceptible to reductive dissolution⁶⁹. Coward et al.⁶⁶ recently proposed several mechanisms by which SRO Fe^{III}-OM phases could become resistant to reductive dissolution, including acquiring reduction-resistant surface coatings, or becoming embedded in a composite aggregate structure⁶. Such a protective coating could even come from higher crystallinity Fe (oxyhydr)oxides, as Hall et al. recently illustrated ¹⁴C-derived C residence time in humid soils was positively correlated with Fe phase crystallinity⁷⁰. Consistent with that, in our work here we found that the 2nd oxidation event generated more crystalline ⁵⁷Fe phases (Table 1) and that the 2nd Fe^{II} oxidation did not stimulate additional C mineralization (Fig.6 and Table 2). It may be that during

repeated redox fluctuations a substantial portion of the co-precipitated OM is lost, but a core Fe-MAOM structure would remain resistant to reductive dissolution.

Perhaps most compelling is the growing evidence that various aggregation, conformation, and structural characteristics of soils confer protection for OM^{5, 6, 7, 10}. Even the protective surface coatings^{66, 70} or conformational changes in OM at low C/Fe ratios⁶⁴ discussed above are examples of micro-aggregate structures not unlike the encasement of SRO Fe-OM phases by aluminosilicate clays or other processes that generate micro-aggregates of minerals and OM during pedogenesis^{6, 7, 10, 71}. These aggregation processes can structure microaggregates with core SRO Fe phases and outer aluminosilicate or other phases that are not susceptible to reductive dissolution—as observed in andisols by dithionite-resistant SRO-Fe phases⁶⁵. Our soil slurry approach was designed to minimize the physical constraints (macro-pore flow, spatial arrangement of microbes, minerals and OM, and the development of aggregates) on C decomposition and thereby isolate the sorbent and electron-transfer roles of Fe in C dynamics (SI Fig. 1). Under these conditions, we find that Fe does not confer intrinsic protection for OM in redox-dynamic soils. In an *in situ* soil environment—where MAOM emerges in a dynamic 3-dimensional space—structural and physical protection of MAOM is thus likely a key protective mechanism for reconciling the comparatively large proportions of SRO-OM associations in soil of very old age based on ¹⁴C-dating^{1, 4, 5, 59}. Future studies should thus assess the extent that the formation and destruction of Fe-cemented microaggregates contribute to OM persistence in redox-dynamic soils. Our work demonstrates that the inherent persistence of SRO Fe-associated C cannot be guaranteed. Biological and geochemical context is critical for understanding the long-term fate of Fe^{III}-associated SOM under a changing climate, given the dual roles of Fe^{III} phases in both accelerating and inhibiting OM decomposition.”

Curiously, the numbers in Fig 1 panel d) appear to be related to the major objective of this manuscript, yet they are not comprehensively discussed. This reviewer found only two out of a total of 8 of these numbers mentioned in the text (+32 % and + 74 % in line 256), without reference to Figure 1.

Response: In the revised manuscript, first we synthesized Fig.1d at the beginning of the “*Results and Discussion*” Section, and also extensively cited Fig.1d as leading results throughout the *Results and Discussion*

(P9-10L194-212) “Consistent with a protective role, under static oxic conditions we found that Fe^{II} oxidation in the presence of added ¹³C-DOM resulted in SRO Fe-C associations that not only inhibited the mineralization of ¹³C-DOM by 35% relative to controls, but also suppressed the priming of native SOM mineralization by 47%, which consequently decreased overall CO₂ production by 33% (Fig.1d). However, when ¹³C-DOM was not added, Fe^{II} oxidation and the production of reactive oxygen species stimulated mineralization of native SOM by 8% relative to the controls (Fig.1d). Thus, the formation of additional SRO-Fe phases did not provide net protection to SOM unless there was additional DOM present. As might be expected, the protective role of Fe was reversible under anoxic conditions. Although CO₂ production from non-Fe amended treatments during the anoxic period was 68–70% lower than in the static oxic treatment (Fig.1d), the *de novo* SRO Fe-MAOM formed via Fe^{II} oxidation was disproportionately vulnerable to subsequent reduction. This consequently stimulated the mineralization of both added ¹³C-DOM and the native SOM by 74% and 32 – 41%, respectively, and thus increased overall CO₂ production by 41 – 49% relative to both non-Fe amended treatments (with or without added DOM, Fig.1d). As a result of Fe-stimulated C mineralization, the anaerobic ¹³C-DOM mineralization was 81% greater than the oxic control. This study highlights that Fe’s electron transfer roles can largely counteract the strong protective effect of SRO Fe^{III}-C associations and sustain C decomposition in redox-dynamic systems.”

RECOMMENDATION

before publication is contemplated, the piece should be revised in two parts:

a) The Introduction must be harmonized such that a clear logical string leading from question asked over measures taken to answer obtained becomes fully visible.

Response: Following the comments from reviewer #1 and 3, we have significantly revised the *Introduction* to follow a clear logical string leading from our objectives. Please see P3-8.

b) The Synthesis part must refer back to the question asked and should be dedicated to a clear indication of the type of success achieved. When the promise was to quantify the relative contributions of different mechanisms, then please discuss the implications of the quantitative relationships found and derive an overall insight that may serve as a take-home message for the reader.

Response: Thank you for the insightful comments. Based on comments from reviewer #1 and 3, we have re-wrote the *Synthesis* section by refereeing back to the question we asked, clarifying the novel findings from our study, and adding discussion on the implications of Fe-C quantitative relationships. Please see P20-23. We believe that after addressing these comments our manuscript is greatly improved. Thank you for your attention to these details.

Reviewer #2 (Remarks to the Author):

Review of submitted Nature Communications manuscript NCOMMS-19-35427, 27
November 2019

Title: Iron-mediated organic matter decomposition can counteract protection

Authors: Chunmei Chen, Steven J. Hall, Elizabeth Coward, and Aaron Thompson

General Comments

The authors present a study on the interactions between iron oxides and organic matter in a series of soil incubations with labelled dissolved organic matter (^{13}C -DOM) and iron (^{57}Fe), both under oxic conditions and under varying redox conditions (oxic-anoxic-oxic). They find that iron oxides protected native soil and freshly added DOM from remineralization when iron and fresh DOM were added together, as reported in other studies, but also that the addition of highly reactive iron oxides alone (formed from the oxidation of FeII under oxic conditions) lead to enhanced degradation of native soil OM under anoxic conditions owing to their role as electron acceptors. They conclude that iron oxides can protect DOM from degradation only when they are themselves protected from reduction by the co-precipitated DOM. I agree that this conclusion is well supported by the data presented in this work.

This is a very well written, high quality paper with an original experimental approach that allows distinguishing between native soil organic carbon and fresh plant-derived organic carbon through ^{13}C labelling, as well as between neo-formed reactive iron oxides also formed in situ and the different forms of iron minerals in the native soil through ^{57}Fe labelling. The data looks of excellent quality if the standard deviations reported in graphs and tables were obtained from real replicates, as I think they were based on the information provided (but see specific comments below). Mössbauer spectroscopy data brings an interesting layer of complementary information on mineral crystallinity

allowing the authors to propose a mechanistic explanation for the bulk and isotopic data derived from the incubation experiment.

My only concerns mostly have to do with additional information that I feel is missing, or explanations that I believe should be provided. None of these concerns are major however, and I believe that the manuscript should be accepted with minor modifications as it meets the high-quality criteria expected from a publication in Nature Communications. This is an important contribution to the field not so much because of the conclusion on the existence of a mutual protection mechanism between DOM and iron oxides, but because of the elegance of the approach taken to demonstrate the validity of this hypothesis.

Response: We appreciate your positive comments very much. They are very encouraging.

Specific comments

1. P. 1, Title: I suggest modifying the title of the manuscript to better reflect the fact that the study focused on soil organic matter: “Iron-mediated organic matter decomposition in soils can counteract protection”.

Response: Change has been made: “Iron-mediated organic matter decomposition in humid soils can counteract protection”.

2. P. 6, line 125: Since the Methods section comes later in the paper, I suggest adding a short contextualization sentence here before immediately jumping to the description of the results.

Response: In the revised manuscript, a short contextualization sentence was added before jumping to the results. Please see *P10L212-215*: “Below we provide details on the

production of the Fe-C MAOM, discuss the data supporting Fe protection of C along with the data supporting Fe stimulation of C loss, and then provide a synthesis of the work.”

3. P. 8, line 171-172: The sorption capacity of about 1.0 (C/Fe ratio), as presented in your reference #41, corresponds to sorption on the surface of preformed FeIII oxides.

Capacities can be higher when FeIII oxides are precipitated from the oxidation of FeII in the presence of DOC, and therefore the ¹³C-DOM remaining in solution might be DOM with a low affinity for the FeIII oxides rather than ¹³C-DOM in excess.

Response: Thanks for the comments. We totally agree. We added a short sentence in P12L269-270: “Co-precipitation could result in Fe-OM associations with much higher C/Fe ratios ^{11, 12, 19}.”

We also stated that ¹³C-DOM remaining in solution might be DOM with a low affinity for the Fe^{III} oxides rather than ¹³C-DOM in excess, please see P13L274-277 “Thus, there was likely ¹³C-DOM with a low affinity for Fe^{III} (oxyhydr)oxides that remained as unprotected ¹³C-DOM in the aqueous phase, which likely led to our observation of significant ¹³C-DOM mineralization even in the presence of newly-formed Fe^{III} (oxyhydr)oxides (Fig.4).”

4. P. 11, lines 235-236: The differences in SOM remineralization between the no DOM/no FeII and the no DOM/with FeII treatments as shown on Figure 6b is extremely small and does not appear statistically significant. The claim of a significant difference between the two treatments seem to stem from the numbers in Table 2, where the SOM-derived CO₂ production numbers seem to have been obtained from the sum of the five measurements carried out during the 11-day period under anoxic conditions. If this is correct, then I doubt that the cumulative uncertainty appearing in Table 2 reflects the propagated uncertainty from the 5 individual measurements, or that the replicate measurements are real replicates (one measurement for each one of the 3 replicate (line 401) incubations rather than three measurements on a single incubation experiment). If they are propagated uncertainties on real replicate measurements, I must congratulate you on the quality of the data acquired in this work; if they are not (here and elsewhere), this should be clarified.

Response: Sorry for the confusion. This misunderstanding is probably due to the scale

issue of y-axis of Figure 6b. We re-plotted Figure 6b and enlarged the anoxic period in the small figure that is inserted. In fact, both figure 6b (temporal SOM-derived CO₂) and Table 2 (cumulative SOM-derived CO₂) showed that during the anoxic phase (gray shaded region), there was an indeed significant difference in SOM-derived CO₂ between no DOM/no Fe (purple line) and no DOM/with Fe (red line) treatments. And all the measurements in this study were on three real replicates (one measurement for each one of the 3 replicates) rather than three measurements on a single incubation experiment.

Figure 6: CO₂ production rates from native soil organic matter (SOM) under redox-fluctuating conditions (b). The grey shaded region represents the anoxic phase of the fluctuating redox treatment.

5. P. 12, line 256: Please provide the uncertainty on these percentages.

Response: The uncertainty is added. Please see P16L365: 32±3% and 74±7%

6. P. 12, lines 263-272: Please comment on the possibility that the difference in chemical composition between the water-extractable soil DOM (native DOM) and the fresh, ¹³C-labelled DOM could have influenced the remineralization results during the anoxic portion of the incubation experiment. Ideally, DOM of similar chemical composition but

different isotopic composition should have been used, although I realize how difficult to obtain these samples would be – maybe water extractable DOM from agricultural soils where C4 v. C3 crops have been grown for a few years?

Response: Yes, we agree that the composition of DOM may have affected the mineralization during the anoxic phase. Ideally, DOM of similar chemical composition but different isotopic composition should have been used. But as the reviewer mentioned, it would be very difficult to obtain this kind of sample: maybe the water extractable DOM from agricultural soils where crops have been grown in chambers with ^{13}C -labeled CO_2 , or where C4 v. C3 crops have been grown for a few years as the reviewer mentioned. We added the discussion in the section of “Results and Discussion” and “Synthesis” in the revised manuscript:

P17-18L373-389 “During the anoxic periods, the mineralization of ^{13}C -DOM over the native SOM (in both the ^{13}C -DOM only and DOM-Fe addition treatments) was 2 – 3 times higher than that in the static oxic treatment at the same point (Fig.4a).

Characterization of the molecular composition of ^{13}C -DOM and water-extractable native SOM using Fourier transform ion cyclotron resonance mass spectrometry (FTICR-MS) revealed that the ^{13}C -DOM had significantly less lignin-derived materials and much more aliphatic formulae than the water-extractable native SOM (Supplementary Table 2 and Fig.4), which represents the most bioavailable fraction of native SOM⁵⁴. The preferential anaerobic mineralization of ^{13}C -DOM over SOM may be due to a lower abundance of lignin-derived compounds, which are not readily depolymerized under anoxic conditions⁵⁵. In addition, compared to water-extractable native SOM, ^{13}C -DOM contains compounds with higher nominal oxidation state of C (NOSC values >0.5 , Supplementary Fig.4), which are associated with a higher likelihood of thermodynamic favorability ($-\Delta G_r$) when coupled to Fe^{III} reduction than the bioavailable fraction of native SOM^{22, 43}.

P21L464-469 “In our study, we even found that the added DOM was preferentially degraded under anoxic conditions relative to the oxic control (Fig.4), which highlights how the thermodynamic constraints of anaerobic metabolism and the molecular composition of C sources can influence the fate of fresh DOM inputs^{22, 42},

⁴³.Consequently, the net effect of Fe-C interactions in dynamic redox environments likely hinges in part on the composition of DOM inputs, a worthy topic for further research.”

7. P. 13, line 293: Please provide the uncertainty on this percentage.

Response: Added. Please see P18L411 “We find that collectively, the reduction of SRO Fe^{III} phases offset O₂ limitations on C mineralization by 24±3% relative to the non-Fe amended treatment (Table 2).”

8. P. 29, Table 1: Are all these iron oxide minerals really “oxyhydr”oxides or simply hydroxides or oxides?

Response: These iron oxide minerals include goethite, lepidocrocite and very SRO-Fe phases that preclude assignment to specific Fe phases from Mössbauer analysis. Goethite and lepidocrocite are oxyhydroxides. We have changed to “Fe^{III} (oxyhydr)oxides” in Table 1 and throughout the manuscript to include all.

9. P. S5, 2nd paragraph: The values measured by ICP-MS for the IRMM-014 standard material are very different from the certified ones, particularly for ⁵⁴Fe (4.56±0.11 vs. 5.84±0.02%, respectively) and ⁵⁶Fe (93.12±0.22 vs. 91.75±0.02%, respectively), reflecting the poor accuracy of isotopes ratios determined by ICP-MS. Did you verify that the offset measured for ⁵⁴Fe and ⁵⁶Fe in pure standard solutions remained the same for the sample solutions with a complex matrix? Please comment.

Response: These average values derived from the repetitive measurements over multiple days (n=3).). The validity of Fe isotope measurements was also tested using mixed solutions of known isotopic ratios. Linear regressions between measured and calculated isotopic ratios of the mixed solution showed R² values of 0.982, 0.987, 0.992, and 0.971 for $f^{54}\text{Fe}$, $f^{56}\text{Fe}$, $f^{57}\text{Fe}$ and $f^{58}\text{Fe}$, respectively (n = 87).

The offset for IRMM-014 was around 1%, which was less than the standard errors (1-7%) associated with the real replicates of samples. Obviously, this ~1% deviation is not appropriate for analyzing differences between samples with natural Fe isotopic

abundance, but should be fine for the ^{57}Fe -spiked samples in this study. We demonstrated this previously for this method and instrument in Tishchenko et al 2015 GCA and see excerpted graph below:

Figure EA1. Precision of $^{57}/^{54}\text{Fe}$ ratio measurements made by single collector ICP-MS (ELAN 6000). Error bars are 2σ based on 30 measurements. The inset illustrates that samples can be statistically distinguished at the 2σ level when they are separated by a $^{57}/^{54}\text{Fe}$ ratio of ~ 0.015

Figure from: Tishchenko, V., C. Meile, M.M. Scherer, T.S. Pasakarnis and A. Thompson, Fe²⁺ catalyzed iron atom exchange and re-crystallization in a tropical soil. *Geochimica et Cosmochimica Acta*, 2015. 148(0): p. 191-202.

10. P. S5, 2nd paragraph: What were the DOC recoveries in the desalination step?

Recoveries can vary a lot from sample to sample when using the PPL cartridges.

Response: We recovered 71% of DOC after desalination and concentration processing, which is higher than the average recovery (62%) reported previously (Dittmar et al. 2008).

This is added in the revised SI P.S5

11. P. S12, 1st paragraph: What was the range of relative intensities of the subtracted background (^{57}Fe signal in the native soils) relative to the intensities of the spikes?

Please provide additional information to allow the reader to appreciate the magnitude of the background correction.

Response: The relative intensities of the subtracted background (^{57}Fe signal in the native soil) only accounted for 8.8 – 18.5% of total ^{57}Fe (spike ^{57}Fe and native soil ^{57}Fe). This information is added in the revised SI P.S12.

Reviewer #3 (Remarks to the Author):

The authors present a paper on the role of synthetic iron-oxides in the decomposition of dissolved organic matter (DOM) in slurries. Experiments were run under controlled redox conditions employing completely mixed batch reactor systems with aqueous suspensions (slurries) that were intensively shaken. The suspensions contained labelled synthetic iron oxides and labeled DOM produced within a labeling experiment with Bermudagrass.

General comments: Abstract, Introduction and synthesis

The authors claim to challenge the OM protection “paradigm”, which was developed originally and further elaborated for soils. The experimental results used to justify their proposition are based on an experimental approach, i.e. completely mixed/shaken batch reactor studies, that does not compare with the “normal” situation of organic matter

storage and sequestration of organic carbon in terrestrial soils. The processes of OM sequestration are quite diverse and incorporate a vast variety of physical, chemical and biological mechanisms that accompany each other during pedogenesis. Of importance and well established is the role of biota (plant roots, earthworms, earth dwelling insects, microbes, fungi). The interplay of the physical, chemical and biological processes results in an accrual of organic matter over time and thus in a positive OC budget for most of the soil orders. Protection from respiratory or reductive degradation, which factual in soils, is mediated by the processes of structure formation which results in an explicit, soil group specific structural arrangement of the components that build up the soil architecture.

Response: We are very grateful to the reviewer for the insightful comments on our manuscript! The objective of this study was to quantify the relative contributions of Fe functional roles in C mineralization. The formation of the organo-mineral associations via sorption, serve as composite building units of the microaggregates, thus the sharp differentiation of OM stabilized chemically by sorption at Fe mineral surfaces and that stabilized physically by occlusion in aggregation is very challenging. Therefore we used a soil slurry approach to minimize the physical-protection mechanisms of aggregation and the spatial separation of decomposers, substrates, and mineral surfaces, and focus on Fe's sorbent and electron-transfer roles in C mineralization. In addition, the slurry approach also facilitates us to homogeneously label natural soil with ^{13}C and ^{57}Fe dual isotopes.

Following the reviewer's comments, we have added the rationale for the soil slurry approach in the *Introduction* section as follows: "In this study, we sought to quantify the relative contributions of Fe functional roles in retarding and accelerating C loss in the initial stages of MAOM formation, where physical constraints (macroaggregation, etc.) on decomposition were minimized (Fig.1)..... We performed the experiments in a soil slurry to minimize the physical-protection mechanisms of aggregation and the spatial separation of decomposers, substrates, and mineral surfaces, and thus focus on Fe's sorbent and electron-transfer roles." P8L164-181

In the *Synthesis* section, we also added discussion about our approach and the implications of our findings in the *in-situ* natural soils, where the interplay of the physical, chemical and biological processes results in an accrual of organic matter over time:

“Perhaps most compelling is the growing evidence that various aggregation, conformation, and structural characteristics of soils confer protection for OM^{5, 6, 7, 10}. Even the protective surface coatings^{66, 70} or conformational changes in OM at low C/Fe ratios⁶⁴ discussed above are examples of micro-aggregate structures not unlike the encasement of SRO Fe-OM phases by aluminosilicate clays or other processes that generate micro-aggregates of minerals and OM during pedogenesis^{6, 7, 10, 71}. These aggregation processes can structure microaggregates with core SRO Fe phases and outer aluminosilicate or other phases that are not susceptible to reductive dissolution—as observed in andisols by dithionite-resistant SRO-Fe phases⁶⁵. Our soil slurry approach was designed to minimize the physical constraints (macro-pore flow, spatial arrangement of microbes, minerals and OM, and the development of aggregates) on C decomposition and thereby isolate the sorbent and electron-transfer roles of Fe in C dynamics (SI Fig. 1). Under these conditions, we find that Fe does not confer intrinsic protection for OM in redox-dynamic soils. In an *in situ* soil environment—where MAOM emerges in a dynamic 3-dimensional space—structural and physical protection of MAOM is thus likely a key protective mechanism for reconciling the comparatively large proportions of SRO-OM associations in soil of very old age based on ¹⁴C-dating^{1, 4, 5, 59}. Future studies should thus assess the extent that the formation and destruction of Fe-cemented microaggregates contribute to OM persistence in redox-dynamic soils. Our work demonstrates that the inherent persistence of SRO Fe-associated C cannot be guaranteed. Biological and geochemical context is critical for understanding the long-term fate of Fe^{III}-associated SOM under a changing climate, given the dual roles of Fe^{III} phases in both accelerating and inhibiting OM decomposition.” P23L494-516

While I strongly respect the research methodology and results obtained, the double labeling approach and the Mössbauer-spectroscopy, I urge the authors to discuss their results in view of pedogenesis by respecting much more the pertinent environmental

conditions and the structural arrangement of components in soil (Already tentatively done so in the synthesis section).

Response: As the reviewer suggested, we have extended the discussion on our results by thinking more about the natural pedogenesis, please P21-23:

“(2) Our work here suggests that, in much the same way wood or steel in a building needs to be painted or encased to protect them from decay and corrosion, the initial SRO Fe-C associations are not likely to persist without protection from periodic Fe reduction events. Several researchers have identified or produced SRO Fe^{III}-OM colloids that are resistant to either microbial or chemical reduction⁶²⁻⁶⁶, however the key components conferring this protection are variable and/or elusive. Some work has identified that SRO Fe^{III}-OM co-precipitates with low C/Fe ratios provide resistance to microbial reduction^{62, 63}, whereas other work has emphasized structural properties (conformation and micro-aggregation) as the mechanism that retards dissolution⁶⁴⁻⁶⁶. SRO Fe-OM phases are often co-precipitated with Al and Si ions⁶⁷—which can retard recrystallization⁶⁸—and given the co-association of Al and Fe with OM in humid soils, Al is a strong candidate for protecting Fe against reduction. However, studies that have examined Al and Si co-precipitated Fe-(oxyhydr)oxides find those ions also make the co-precipitates more susceptible to reductive dissolution⁶⁹. Coward et al.⁶⁶ recently proposed several mechanisms by which SRO Fe^{III}-OM phases could become resistant to reductive dissolution, including acquiring reduction-resistant surface coatings, or becoming embedded in a composite aggregate structure⁶. Such a protective coating could even come from higher crystallinity Fe (oxyhydr)oxides, as Hall et al. recently illustrated ¹⁴C-derived C residence time in humid soils was positively correlated with Fe phase crystallinity⁷⁰. Consistent with that, in our work here we found that the 2nd oxidation event generated more crystalline ⁵⁷Fe phases (Table 1) and that the 2nd Fe^{II} oxidation did not stimulate additional C mineralization (Fig.6 and Table 2). It may be that during repeated redox fluctuations a substantial portion of the co-precipitated OM is lost, but a core Fe-MAOM structure would remain resistant to reductive dissolution.

Perhaps most compelling is the growing evidence that various aggregation, conformation, and structural characteristics of soils confer protection for OM^{5, 6, 7, 10}.

Even the protective surface coatings^{66, 70} or conformational changes in OM at low C/Fe ratios⁶⁴ discussed above are examples of micro-aggregate structures not unlike the encasement of SRO Fe-OM phases by aluminosilicate clays or other processes that generate micro-aggregates of minerals and OM during pedogenesis^{6, 7, 10, 71}. These aggregation processes can structure microaggregates with core SRO Fe phases and outer aluminosilicate or other phases that are not susceptible to reductive dissolution—as observed in andisols by dithionite-resistant SRO-Fe phases⁶⁵. Our soil slurry approach was designed to minimize the physical constraints (macro-pore flow, spatial arrangement of microbes, minerals and OM, and the development of aggregates) on C decomposition and thereby isolate the sorbent and electron-transfer roles of Fe in C dynamics (SI Fig. 1). Under these conditions, we find that Fe does not confer intrinsic protection for OM in redox-dynamic soils. In an *in situ* soil environment—where MAOM emerges in a dynamic 3-dimensional space—structural and physical protection of MAOM is thus likely a key protective mechanism for reconciling the comparatively large proportions of SRO-OM associations in soil of very old age based on ¹⁴C-dating^{1, 4, 5, 59}. Future studies should thus assess the extent that the formation and destruction of Fe-cemented microaggregates contribute to OM persistence in redox-dynamic soils. Our work demonstrates that the inherent persistence of SRO Fe-associated C cannot be guaranteed. Biological and geochemical context is critical for understanding the long-term fate of Fe^{III}-associated SOM under a changing climate, given the dual roles of Fe^{III} phases in both accelerating and inhibiting OM decomposition.”

In addition, the authors should discuss their experimental approach (Adding of DOM and Fe in a slurry) in view of the redox-driven Fe-dynamics in soil: What would be the source of reduced iron required to allow for the formation of Fe-OM in the natural situation where iron is not added but replenished by weathering of Fe-bearing minerals? What about the action of soil biota?

Response: Very good comments. Based on the reviewer’s comments, first we added the texts in the *Introduction* section to explain the formation of SRO Fe-C associations in the

pertinent redox-dynamic environments as a result of Fe^{II} oxidation in the presence of DOM, please see P4-5L87-105:

“Soils form as primary minerals are weathered to secondary minerals and organic materials are incorporated, degraded and transformed to biomass and necromass by the microbial community. A commonly accepted mechanism for the initiation of MAOM is for DOM of plant or microbial origin¹⁶ to be sequestered via sorption or co-precipitation reactions with existing and newly formed minerals^{5, 17-19}. One particularly important mechanism for formation of Fe-associated OM involves the oxidation of Fe^{II} to Fe^{III} at redox interfaces and its rapid hydrolysis to Fe^{III} (oxyhydr)oxides, which co-precipitate with DOM in the soil solution²⁰. This can occur wherever Fe^{II}-bearing anoxic solutions come in contact with O₂, such as in periodically flooded soil horizons or across redox gradients that often occur within soil aggregates in upland soils under humid climates²⁰⁻²². High rates of Fe reduction have been observed in surface soils during periods of elevated moisture and high biological activity, leading to a heterogeneous distribution of iron within soil profiles²³⁻²⁶. Iron reduction appears to be a ubiquitous soil biogeochemical process across a broad range of terrestrial ecosystems²³⁻³⁰. Across these ecosystems, C:Fe molar ratios of Fe-C associations point to the dominance of co-precipitation vs. adsorption^{11, 12}. These lines of evidence place the epicenter of Fe-associated C formation at these dynamic anoxic-oxic interfaces in surface soils.”

In addition, we also added discussion in the *Synthesis* section to clarify the relevance of our study with the natural environments: “Humid soils are replete with microsites that undergo dynamic anoxia in response to high labile C loads during periods of high moisture and experience appreciable Fe^{III} reduction rates^{23, 25, 60, 61}. Oxidation of the Fe^{II} generated from Fe^{III} reduction is a common mechanism for MAOM formation in humid soils, yet Fe is also responsive for OM loss in humid soils and our work here illustrates two principal refinements in this regard.”P20-21L450-455

SRO phases of Iron, Aluminum, and Manganese newly formed in the soil environment are protected by being incorporated in the soils aggregate systems, here predominantly as

organo-mineral or mineral-organic associations within (micro)aggregates. While the existence of the hierarchy of a system of aggregates is still in debate, aggregation and soils aggregated structure is not disputed. It is also well established that SRO-OM associations are part of the aggregated structure of soils. Aggregation and the emerging architecture of soils is the searched for feature that provides the protection. Or, to state this in alternative phrase – already given by the authors but as a conclusive remark: The processes of soil aggregation provide the protection not only of the SRO Fe phases, but of the associations build of SRO-Fe/Al/Mn and organic matter.

From the results obtained by the authors I would rather conclude that the fact that we find comparatively large proportions of SRO-OM associations in soil of very old age (based on C14-dating of the OM of the associations) strengthens the concept of protection by aggregation. Given the findings of the authors and others on the sensitiveness of SRO-Fe to a rather rapid decomposition in the absence Oxygen or reductive conditions, the already proposed process of protection in soil aggregates puts the SRO-Fe-OM stabilization paradigm in a greater framework.

Response: We agree with reviewer that the processes of soil aggregation provide the protection not only of the SRO Fe phases, but of the associations built of SRO-Fe and organic matter. Our findings may further strengthen the concept of SRO Fe-C protection by aggregation. We added one paragraph in the *Synthesis* section to discuss that soil aggregation may put the SRO-Fe-OM stabilization paradigm in a greater framework. Please see P22-23L494-510:

“Perhaps most compelling is the growing evidence that various aggregation, conformation, and structural characteristics of soils confer protection for OM^{5, 6, 7, 10}. Even the protective surface coatings^{66, 70} or conformational changes in OM at low C/Fe ratios⁶⁴ discussed above are examples of micro-aggregate structures not unlike the encasement of SRO Fe-OM phases by aluminosilicate clays or other processes that generate micro-aggregates of minerals and OM during pedogenesis^{6, 7, 10, 71}. These aggregation processes can structure microaggregates with core SRO Fe phases and outer aluminosilicate or other phases that are not susceptible to reductive dissolution—as observed in andisols by dithionite-resistant SRO-Fe phases⁶⁵. Our soil slurry approach was designed to minimize the physical constraints (macro-pore flow, spatial arrangement

of microbes, minerals and OM, and the development of aggregates) on C decomposition and thereby isolate the sorbent and electron-transfer roles of Fe in C dynamics (SI Fig. 1). Under these conditions, we find that Fe does not confer intrinsic protection for OM in redox-dynamic soils. In an *in situ* soil environment—where MAOM emerges in a dynamic 3-dimensional space—structural and physical protection of MAOM is thus likely a key protective mechanism for reconciling the comparatively large proportions of SRO-OM associations in soil of very old age based on ^{14}C -dating^{1, 4, 5, 59}.”

Specific comments:

Adding Fe to anoxic conditions may also result, after reduction to ferrous iron, in the formation of Fe-complexes with low-molecular weight DOM. Whether or not this may occur in the experiments and to what extent the results are affected remains unconsidered. Please comment.

Response: We agree that -DOM may interact with Fe^{II} to form DOM- Fe^{II} complexes. Our data indeed showed that “The treatment with both ^{57}Fe and ^{13}C -DOM added had 10% more adsorbed $^{57}\text{Fe}^{\text{II}}$ than the $^{57}\text{Fe}^{\text{II}}$ -only treatment before oxidation (Fig.2a and 2b), likely due to co-sorption of the Fe^{2+} -DOM complex, as observed previously⁴⁴.”
(P10L221-222)

However, following exposure to O_2 , all Fe^{II} was completely oxidized with or without DOM. One important finding from this study was that the addition of ^{57}Fe and ^{13}C -DOM together resulted in the formation of even lower crystallinity SRO Fe^{III} (oxyhydr)oxides than the ^{57}Fe addition-only treatment as illustrated by the lower 35K/5K and 12K/5K crystallinity ratios (Table 1; Supplementary Fig.7). The suppression of the crystallinity by DOM may be hinged in part on the co-sorption of Fe^{2+} -DOM complexes.

L49: There is ample evidence that also secondary clay minerals formed during pedogenesis are important for the accrual of OM in soil. Such secondary clay minerals are not considered in the study of Rasmussen et al. 2018. The base their statement of the “clay content” which is the operational fraction of soil components that are smaller than $<2\mu\text{m}$!

Response: Based on reviewer's comments, we changed this sentence to "A growing body of studies has demonstrated that geochemical factors, such as secondary clay minerals and short-range-ordered (SRO) iron (Fe) and aluminum (Al) phases in particular, are vital determinants of C accrual¹⁻³.", please check P3L53-55

L59-60: The consequence of this "structural role" is the protection of OM against decomposition. It is this structural role that is the basis of the paradigm of OM protection by pedogenic oxides. This should must be emphasized and discussed in the frame of the study.

Response: Thanks for the comments. We have emphasized and discussed the consequence of the "structural role" in OM protection in the revised section of *Introduction*. Please see P3-4L54-71:

"Mineral-associated organic matter (MAOM) is thought to persist because organic matter (OM) can form strong chemical bonds to minerals and can be physically protected in microaggregates or co-precipitates^{4,5}. Once the initial association of OM with minerals has occurred, soil structural conditions (aggregate formation, macro-scale shifts in fluid flowpaths, etc.) can further isolate and compartmentalize OM substrates from decomposer organisms and restrict the diffusion of oxygen, thus further protecting soil organic matter (SOM) against decomposition^{6,7}. These features can potentially lead to longer turnover times for MAOM than for particulate organic matter^{8,9}, and may explain MAOM residence times of centuries–millennia^{4,5,10}. A large portion of MAOM in both soils and sediments is adsorbed or co-precipitated with Fe minerals¹¹⁻¹³, and Fe (oxyhydr)oxides may be among the most important drivers of organo-mineral associations in subsoils⁴."

L61-62: Most of the studies so far concentrated on experimental approaches focusing on sorption/coprecipitation from suspensions/slurries including the work of the authors. A "paradigm" challenging approach should explore more realistic conditions typical for soils. Soils are not slurries or suspensions.

Response: Please see our response to Reviewer's first comment. The objective of this study was to quantify the relative contributions of Fe functional roles in C mineralization. The formation of the organo-mineral associations via sorption, serve as composite building units of the microaggregates, thus the sharp differentiation of OM stabilized chemically by sorption at Fe mineral surfaces and that stabilized physically by occlusion in aggregation is very challenging. Therefore we used a soil slurry approach to minimize the physical-protection mechanisms of aggregation and the spatial separation of decomposers, substrates, and mineral surfaces, and focus on Fe's sorbent and electron-transfer roles in C mineralization. In addition, the slurry approach also facilitates us to homogenously label natural soil with ^{13}C and ^{57}Fe dual isotopes.

Following the reviewer's comments, we have added the rationale for the soil slurry approach in the *Introduction* section as follows: "In this study, we sought to quantify the relative contributions of Fe functional roles in retarding and accelerating C loss in the initial stages of MAOM formation, where physical constraints (macroaggregation, etc.) on decomposition were minimized (Fig.1)..... We performed the experiments in a soil slurry to minimize the physical-protection mechanisms of aggregation and the spatial separation of decomposers, substrates, and mineral surfaces, and thus focus on Fe's sorbent and electron-transfer roles." P8L164-180

In the *Synthesis* section, we also added discussion about our approach and the implications of our findings in the *in-situ* natural soils, where the interplay of the physical, chemical and biological processes results in an accrual of organic matter over time: "Perhaps most compelling is the growing evidence that various aggregation, conformation, and structural characteristics of soils confer protection for OM^{5, 6, 7, 10}. Even the protective surface coatings^{66, 70} or conformational changes in OM at low C/Fe ratios⁶⁴ discussed above are examples of micro-aggregate structures not unlike the encasement of SRO Fe-OM phases by aluminosilicate clays or other processes that generate micro-aggregates of minerals and OM during pedogenesis^{6, 7, 10, 71}. These aggregation processes can structure microaggregates with core SRO Fe phases and outer aluminosilicate or other phases that are not susceptible to reductive dissolution—as observed in andisols by dithionite-resistant SRO-Fe phases⁶⁵. Our soil slurry approach

was designed to minimize the physical constraints (macro-pore flow, spatial arrangement of microbes, minerals and OM, and the development of aggregates) on C decomposition and thereby isolate the sorbent and electron-transfer roles of Fe in C dynamics (SI Fig. 1). Under these conditions, we find that Fe does not confer intrinsic protection for OM in redox-dynamic soils. In an *in situ* soil environment—where MAOM emerges in a dynamic 3-dimensional space—structural and physical protection of MAOM is thus likely a key protective mechanism for reconciling the comparatively large proportions of SRO-OM associations in soil of very old age based on ¹⁴C-dating^{1, 4, 5, 59}. Future studies should thus assess the extent that the formation and destruction of Fe-cemented microaggregates contribute to OM persistence in redox-dynamic soils. Our work demonstrates that the inherent persistence of SRO Fe-associated C cannot be guaranteed. Biological and geochemical context is critical for understanding the long-term fate of Fe^{III}-associated SOM under a changing climate, given the dual roles of Fe^{III} phases in both accelerating and inhibiting OM decomposition.” P23L494-516

L75: “ubiquitous” is an overstatement. Based on the two given references, it is not possible to elude on the “ubiquitomes” of anoxic microsites in soils. While there is no doubt on the existence of anoxic sites, there is no systematic study on the proportion and persistence of such sites in terrestrial soil orders. In contrast to soils with hydromorphic features (semi-terrestrial soil orders), anoxic conditions are limited to the interior of aggregates that support the respective water retention characteristics/hydraulic properties.

Response: Thanks for pointing this out. To avoid the overstatement, in the revised manuscript we have changed this sentence to: “High rates of Fe reduction have been observed in surface soils during periods of elevated moisture and high biological activity, leading to a heterogeneous distribution of iron within soil profiles²³⁻²⁶. Iron reduction appears to be a ubiquitous soil biogeochemical process across a broad range of terrestrial ecosystems²³⁻³⁰.”P5L97-101

L78-80. There are several studies that specifically explore the reduction/decomposition of Fe-OM and report on the impact on the release of both Fe and DOM as a function.

Response: To avoid confusion, this sentence has been changed to “Thus, while redox-cycling of Fe in soils generates SRO Fe^{III}-OM associations that likely promote OM persistence (sorbent and structural roles of Fe), it also promotes OM decomposition (electron transfer roles of Fe) through Fenton chemistry during Fe^{II} oxidation and as a consequence of Fe^{III} reduction reactions during periodic anoxia.” (P6L132-136)

L85-87: Destruction of the Fe-cemented aggregates by biologic processes and concomitant degradation of OM is one of the processes presumable counteracting OM protection. A better understanding of the mechanisms behind these processes under the conditions met would much better help to test the Fe-OM-stabilization paradigm.

Response: we agree that destruction of the Fe-cemented aggregates can also counteract OM protection. However our experimental design is already quite complicated and assessing the role (de)aggregation processes in OM mineralization will add more complexity. In addition, our objective was to quantify the relative contributions of Fe functional roles in C mineralization. However, it is impossible for us to quantitatively differentiate Fe's structural and sorbent roles. Therefore we choose to focus on the sorbent and electron transfer roles in this study. But we recommend that “Future studies should thus assess the extent that the formation and destruction of Fe-cemented microaggregates contribute to OM persistence in redox-dynamic soils.” P23L510-512”

L99-110: This is a truncated and oversimplified restitution of the “paradigm”. The full story reads that secondary SRO-Fe/Al-phases show a high affinity for DOM and form associations in soil. However, it is also recognized that they serve as a good substrate and electron donor/acceptor source for microbes and will decompose if not other processes provide protection. Numerous studies showed this under various oxic and anoxic conditions using batch and column techniques. The missing part to understand protection is structure formation and aggregation. Aggregation is the major pathway that results in (physical) protection. The authors should take this part of the “paradigm” into account and revise the MS in accordance.

Response: Thank you for your suggestions. By following reviewer 1 and 3's comments, we elaborate this "paradigm" by taking aggregation into account and also clearly clarified a knowledge gap that needed to be addressed. Although it has been shown that Fe oxidation and reduction can result in C loss and preservation, these processes are rarely explored concurrently such that their relative impacts on soil C cycling remain unclear.

Please see P7L140-154:

"Despite evidence for the importance of Fe-stimulated decomposition, the dominant paradigm in Earth System science has focused on Fe-mediated OM protection via adsorption, co-precipitation or aggregation^{5, 7, 12, 19, 20, 37-40}. While it is also recognized that Fe-OM associations are formed during redox-cycling of Fe, and that both oxidation and reduction of Fe can lead to C loss and mineralization^{31-33, 36}, these processes are rarely explored concurrently. In fact, only a few studies have directly measured the microbial availability of Fe-associated OM in soils^{40, 41}, and studies that highlight Fe-associated C in anoxic or redox-interface sediments do not explore why these Fe^{III} minerals persist despite being thermodynamically poised for reductive dissolution^{12, 20}—this topic is instead discussed in separate studies highlighting that Fe mineral stability in anoxic zones derives from thermodynamic constraints on decomposition of OM substrates^{22, 42, 43}. Examining these competing functional roles together remains a critical knowledge gap for understanding the persistence or potential losses of Fe-associated C under climate and land-use change."

L117-119: Dual-isotope utilization is a very appropriate approach for such a study. I liked this very much!

Response: Many thanks for your positive comments

L144: The slurry does not resemble the situation of a "complex natural soil". Rephrase.

Response: In the revised manuscript, we rephrased the "complex natural soil" to "a complex soil system containing a mixture of aluminosilicates, Fe^{III} (oxyhydr)oxides, and a variety of organic compounds". Please see P11L237-238.

L180: Why do you classify carbon input by soil biota as “exogenous”.

Response: To avoid the confusion, this sentence has been changed to “This statement has been changed to “Labile C inputs are often observed to alter the decomposition of extant SOM, defined as priming^{47, 48}”, please check P23L281-282

L251, 284: Have you considered the formation of complexes build from ferrous iron and DOM? Such mechanisms have been reported in the literature and may have to be considered in the experimental outcomes. In a natural soil, such complexes would be eventually prone to export with seepage in humid climates. Please comment.

Response: We agree that -DOM may interact with Fe^{II} to form DOM-Fe^{II} complexes. Our data indeed showed that “The treatment with both ⁵⁷Fe and ¹³C-DOM added had 10% more adsorbed ⁵⁷Fe^{II} than the ⁵⁷Fe^{II}-only treatment before oxidation (Fig.2a and 2b), likely due to co-sorption of the Fe²⁺-DOM complex, as observed previously⁴⁴.” (P10L221-222)

However, following exposure to O₂, all Fe^{II} was completely oxidized with or without DOM. One important finding from this study was that the addition of ⁵⁷Fe and ¹³C-DOM together resulted in the formation of even lower crystallinity SRO Fe^{III} (oxyhydr)oxides than the ⁵⁷Fe addition-only treatment as illustrated by the lower 35K/5K and 12K/5K crystallinity ratios (Table 1; Supplementary Fig.7). The suppression of the crystallinity of Fe^{III} (oxyhydr)oxides by DOM may be hinged in part on the co-sorption of Fe²⁺-DOM complexes.

We agree that a portion of Fe²⁺-DOM complex could be exported with seepage in humid climates, but this experiment is a closed reaction system.

Methods

L332ff: Why did you use Bermudagrass for the labeling experiment? How does Bermudagrass compare to flora of the agricultural sites used for sampling the cultivated soil (L354-355). Is there a bias in OM quality to be expected?

Response: Thanks for the comments. The flora in this site includes hay and a few annual crops (e.g., *Zea mais*, *Triticum aestivum*). We added a sentence to explain why we used Bermudagrass for the labeling experiment in the section of *Methods*, please see P23 L520-523: “DOM is inherently heterogeneous, diverse and dynamic in composition⁶, and here we used bermudagrass-extracted DOM to encompass a mixture of organic molecules representative of those that derive from early stage herbaceous litter decomposition.”

In our study, we even found that the added DOM was preferentially degraded under anoxic conditions relative to the oxic control (Fig.4), which highlights how the thermodynamic constraints of anaerobic metabolism and the molecular composition of C sources can influence the fate of fresh DOM inputs^{22, 42, 43}. Consequently, the net effect of Fe-C interactions in dynamic redox environments likely hinges in part on the composition of DOM inputs, a worthy topic for further research.

L339ff: Based on the FTICR-MS: How does the quality of the bermudagrass OM compare to natural DOM <0,2µm? Give details

Response: This information is added in the revised section of *Methods*: “Compared to water-extractable natural SOM, the bermudagrass-derived DOM had significantly more aliphatic compounds with less lignin-derived materials”. Please see P24L535-537.

L359: More details on the soil materials used and the soil used should be given. The Redoximorphic features would much more point to the direction of adults? Ultisols have a broad range of properties, usually strongly acidic and rich in hardly dissolvable Fe-Ox. Cultivation requires melioration of nutrients and H by adding fertilizer and lime. This, of course will affect the type and conditions of the linkage of Fe-OM.

Response: Thanks for asking. The details on the soil materials including soil series, fertilizer, liming and redox-history have been provided in the revised section of *Methods*. Please see P25L551-562:

“The soil used is classified as fine kaolinitic, thermic Typic Kanhapludults of the Appling series, derived from granitic gneiss. We collected soils from cultivated land on an interfluvial managed for hay and a few annual crops (e.g., *Zea mais*, *Triticum aestivum*).

Current management practice includes annual plowing and disking, the addition of ~4 Mg ha⁻¹ of lime in the last eight years and fertilization of NPK at the rate of 160, 40, and 70 kg ha⁻¹ yr⁻¹, respectively⁷². Interfluvial soils across the Calhoun CZO are characterized by deep soils with pronounced subsurface redoximorphic features^{23, 73} and seasonal fluctuations in Fe reduction events corresponding with antecedent moisture and labile organic C⁷⁴. During the early spring, surface soils in particular experience a peak in Fe^{II} associated with Fe reduction, which subsequently subsides as the soils become drier later in the spring/summer⁷⁴.”

L359ff: Soil material was collected from the first 0-20 cm (topsoil). Are these topsoil horizons known for expressing redoximorphic features?

Response: The surface soil does not display redoximorphic features, but surface soils experience seasonal fluctuations in Fe reduction events corresponding with antecedent moisture and labile organic C⁷⁴. During the early spring, surface soils in particular experience a peak in ferrous iron associated with Fe reduction, which subsequently subsides as the soils become drier later in the spring/summer⁷⁴.

Across Calhoun CZO, Surface soils have much higher Fe^{II} concentration than subsurface soils, although redoximorphic features are often shown in the subsurface soils. We suspect that visual evidence of Fe redox cycling in surface soils may have been obscured by high organic matter content, which shows a dark color. Surface soils typically contain the highest stocks of root biomass and organic matter across a broad range of terrestrial ecosystems. As a consequence of abundant C inputs that generate anaerobic microsites yet obscure the visual effects of Fe reduction, we suggest that cryptic Fe reduction in terrestrial surface soils may be a more commonplace phenomenon than is implied by visible redoximorphic features.

L394ff: Rotary and end-over-end shaking will further destroy aggregates by mechanical stress. How can you exclude that the results are to a larger extent affected by shaking?

Response: In this study, we sought to use gentle shaking to minimize the physical constraints (macro-pore flow, spatial arrangement of microbes, minerals and OM, and the development of aggregates) on C decomposition. In addition, all the treatments were exposed to a same degree of shaking, and therefore shaking should not affect our results from comparing the difference between Fe-amended treatments (with or without DOM) and controls.

L439ff: Henry's law is not directly applicable to slurries. Did you correct for that?

Response: Concentrations of CO₂ were corrected for dissolved gases based on temperature and pH-dependent Henry's law constants (Groffman et al. 1999; Sander, 2015). In addition, we used the actual water volume (not the water + soil volume) for the correction.

Groffman PM., Holland EA, Myrold DD, Robertson PG, Zou XM. Denitrification. Robertson, G. Philip (Editor); Coleman, David C. (Editor); Bledsoe, Caroline (Editor). Standard Soil Methods for Long-Term Ecological Research. Cary, NC, USA: Oxford University Press, 1999. p 272

Sander R. Compilation of Henry's law constants (version 4.0) for water as solvent. Atmospheric Chemistry and Physics, 2015, 15(8):4399-4981.

REVIEWERS' COMMENTS:

Reviewer #2 (Remarks to the Author):

The authors have made an excellent job of addressing all concerns related to the initial submission. I recommend publication of this new version as is.

Reviewer #3 (Remarks to the Author):

The authors present a revised version of the manuscript on the role of synthetic iron-oxides in the decomposition of dissolved organic matter (DOM) in slurries. The revision covers all of my concerns and remarks. While several my original concerns have been properly answered or specified with the revision, I still ask the authors to clarify/revise the following.

1. Explaining in more depth the experimental approach and the focus of the study does not fix the fundamental problem that rather important soil processes are ignored. This, again, asks for a weakening or even a removal of the inference and conclusions drawn for soils. The presented study is interesting providing results worth publishing, however, the transfer to natural soil systems is conditional.
2. Please rule out "paradigm" and in particular "dominant paradigm" and replace by a more appropriate term, e.g., "conception". "Paradigm" is a very general and demanding generic term in the philosophy of science. The notion that OM bound to Fe-phases is the major process for the stabilization of OM against degradation has to be taken much more a perception of a group of soil chemists rather than a paradigm of "earth system science".
3. I am still not convinced that the doubtless interesting results obtained from the "slurry experiment" can be transferred to the understanding of the stabilization of organic matter even in topsoils of humid soils. It would rather require a comparison with results obtained from field observations directly dedicated to (dis)prove this. Therefore, I suggest to weaken statements on the transferability to soil or implications for soil environments. We simply still lack such dedicated field studies.

Specific comments:

Some sentences of the revised version are much too long, e.g. E.g., L97-101, L147-151. Should be shortened by, e.g., splitting in two sentences.

In the discussion, the manuscript would definitely profit from considering research that studied such processes under flow conditions, both in the field and in the lab.

L26 and following: The study will not allow to answer this two-part question, as the experimental approach will only allow for the first part. Rephrase.

L39-40: Delete "as assumed under the current paradigm"

L40: There is no need to "contend": the physicochemical and physical protection is rather well accepted based on the evidence provided in the meantime (see, e.g., lines 51-53).

L71-73: Give reference or delete. There are studies around that attack combinations of at least two functions.

L110,111: Delete statements in brackets.

L114-116: delete "dominant paradigm in Earth System science" and revise.

L128: delete "sought to"

L134: replace "soils" by soil materials.

L137: replace "soils" by soil materials.

L274: Give full name for TPA.

Synthesis: Replace "dovetails"

L353: The citation must be made more cleared and discussed: Among the factors tested by Rasmussen et al, the SRO were found to be the best predictor of C-content. Yet, the selection was

based on data availability, not so much on the current understanding. This is a general disadvantage of statistical explorations. Please revise.

L360 and following: The authors must clarify what soil orders their results should apply for: Upland soils (large number of citations comes from studies using these soils), humid soils, wetland or floodplain soils. As the properties and processes of these soils will strongly differ, they may not fall in a single class with regard to their functioning as soil

L378-380: This metaphor is invalid. Neither by scale nor by composition and structure soils are like "buildings". Delete.

Reviewer #3 (Remarks to the Author):

The authors present a revised version of the manuscript on the role of synthetic iron-oxides in the decomposition of dissolved organic matter (DOM) in slurries. The revision covers all of my concerns and remarks. While several of my original concerns have been properly answered or specified with the revision, I still ask the authors to clarify/revise the following.

1. Explaining in more depth the experimental approach and the focus of the study does not fix the fundamental problem that rather important soil processes are ignored. This, again, asks for a weakening or even a removal of the inference and conclusions drawn for soils. The presented study is interesting providing results worth publishing, however, the transfer to natural soil systems is conditional.

Response: We have revised the text in several places to emphasize that these are soil slurries. Also, we add an explicit call-out in the synthesis section that these results are tentative for field soils.

“The magnitude of these counteracting mechanisms may also be influenced by soil structure, which we eliminated in our study by conducting experiments in soil slurries. Hence, direct application of our results to *in situ* soil environments is tentative. However, the general principles of our work are also likely to be applicable to structurally complex soil systems.”

2. Please rule out “paradigm” and in particular “dominant paradigm” and replace by a more appropriate term, e.g., “conception”. “Paradigm” is a very general and demanding generic term in the philosophy of science. The notion that OM bound to Fe-phases is the major process for the stabilization of OM against degradation has to be taken much more as a perception of a group of soil chemists rather than a paradigm of “earth system science”.

Response: Thanks for your comments. Changes have been made: “paradigm” has been deleted or replaced by “perception”

3. I am still not convinced that the doubtless interesting results obtained from the “slurry experiment” can be transferred to the understanding of the stabilization of organic matter even in topsoils of humid soils. It would rather require a comparison with results obtained from field observations directly dedicated to (dis)prove this. Therefore, I suggest to weaken statements on the transferability to soil or implications for soil environments. We simply still lack such dedicated field studies.

Response: As mentioned in response to #1 above, we add an explicit call-out in the synthesis section that these results are tentative for field soils.

Specific comments:

Some sentences of the revised version are much too long, e.g. E.g., L97-101, L147-151. Should be shortened by, e.g., splitting in two sentences.

Response: we respectfully disagree with the reviewer. In both of these sentences leaving them as written is more fluid and consistent with what we want to convey.

In the discussion, the manuscript would definitely profit from considering research that studied such processes under flow conditions, both in the field and in the lab.

We cite Hagedorn et al. 2000 as one example of a study that assessed Fe-C interactions in soil profiles with flowing water. We are not aware, however, of studies that explicitly targeted multiple functional roles of Fe-C interactions in field settings. We do discuss some research that examines similar processes in the laboratory, by examining soil samples with intact structure. See Hall et al SBB 2018 and Coward et al GCA 2018 and well as Filimonova et al GCA 2016 as examples. These studies point to the importance of soil structure or other processes in protecting reactive (SRO) Fe bound OM.

L26 and following: The study will not allow to answer this two-part question, as the experimental approach will only allow for the first part. Rephrase.

Response: this sentence has been deleted.

L39-40: Delete “as assumed under the current paradigm”

Response: Deleted

L40: There is no need to “contend”: the physicochemical and physical protection is rather well accepted based on the evidence provided in the meantime (see, e.g., lines 51-53).

Response: We have deleted “contend” and change this statement to “Rather, SRO Fe phases require their own physiochemical protection to contribute to OM persistence.”

L71-73: Give reference or delete. There are studies around that attack combinations of at least two functions.

Response: We have deleted the statement that “These Fe functional roles have largely been studied in isolation in previous work”.

L110, 111: Delete statements in brackets.

Response: Deleted

L114-116: delete “dominant paradigm in Earth System science” and revise.

Deleted. Revision has been made:

“the common perception of iron’s role in SOM has focused on Fe-mediated OM protection via adsorption, co-precipitation, or aggregation”

L128: delete “sought to”

Response: Deleted

L134: replace “soils” by soil materials.

Response: Revised to say ‘soil slurries’

L137: replace “soils” by soil materials.

Response: Revised to say ‘soil slurries’

L274: Give full name for TPA.

Response: Given: terephthalate

Synthesis: Replace “dovetails”

Response: Replaced by “is consistent”

L353: The citation must be made more cleared and discussed: Among the factors tested by Rasmussen et al, the SRO were found to be the best predictor of C-content. Yet, the selection was based on data availability, not so much on the current understanding. This is a general disadvantage of statistical explorations. Please revise.

Response: Revised.

“A recent survey of over 5,500 soil profiles spanning continental scale environmental gradients found that SRO Fe and Al (oxyhydr)oxide abundance was the best predictor of C content in humid soils, among the geochemical and climate variables that were available⁴⁵.”

L360 and following: The authors must clarify what soil orders their results should apply for: Upland soils (large number of citations comes from studies using these soils), humid soils, wetland or floodplain soils. As the properties and processes of these soils will strongly differ, they may not fall in a single class with regard to their functioning as soil.

Response: Our thought is that these results would be relevant to any soils that experience periodic anoxia. Although this likely affects most upland soils except those in very arid environments, it most certainly affects those from humid regions. Thus, we state this clearly many times in the manuscript. In response to this comment, we also expand this concept in the text to redox-dynamic soils in general, indicating that would include floodplain soils and perennial wetland soils as well.

L378-380: This metaphor is invalid. Neither by scale nor by composition and structure soils are like “buildings”. Delete.

Response: This metaphor has been deleted.